# Translational control of one-carbon metabolism underpins ribosomal protein phenotypes in cell division and longevity

**Nairita Maitra[1], Chong He[2], Heidi M Blank[1], Mitsuhiro Tsuchiya[3], Birgit Schilling[2], Matt Kaeberlein[3], Rodolfo Aramayo[4], Brian K Kennedy[2,5,6]\*, Michael Polymenis[1]\***

[1]Department of Biochemistry and Biophysics, Texas A&M University, College Station, United States; [2]Buck Institute for Research on Aging, Novato, United States; [3]Department of Pathology, University of Washington, Seattle, United States; [4]Department of Biology, Texas A&M University, College Station, United States; [5]Departments of Biochemistry and Physiology, Yong Loo Lin School of Medicine, National University of Singapore, Singapore, Singapore; [6]Centre for Healthy Ageing, National University of Singapore, National University Health System, Singapore, Singapore

**Abstract** A long-standing problem is how cells that lack one of the highly similar ribosomal proteins (RPs) often display distinct phenotypes. Yeast and other organisms live longer when they lack specific ribosomal proteins, especially of the large 60S subunit of the ribosome. However, longevity is neither associated with the generation time of RP deletion mutants nor with bulk inhibition of protein synthesis. Here, we queried actively dividing RP mutants through the cell cycle. Our data link transcriptional, translational, and metabolic changes to phenotypes associated with the loss of paralogous RPs. We uncovered translational control of transcripts encoding enzymes of methionine and serine metabolism, which are part of one-carbon (1C) pathways. Cells lacking Rpl22Ap, which are long-lived, have lower levels of metabolites associated with 1C metabolism. Loss of 1C enzymes increased the longevity of wild type cells. 1C pathways exist in all organisms and targeting the relevant enzymes could represent longevity interventions.

**\*For correspondence:**
bkennedy@nus.edu.sg (BKK);
polymenis@tamu.edu (MP)

## Introduction

Mutations in ribosomal proteins (RPs) often have distinct phenotypes and, in people, they lead to diseases called ribosomopathies (*Aspesi and Ellis, 2019*; *De Keersmaecker et al., 2015*; *Goudarzi and Lindström, 2016*; *Mills and Green, 2017*). How perturbations of a general process such as protein synthesis can lead to specificity in the observed phenotypes is puzzling. Specialized ribosomes with different composition may account for the distinct phenotypes of ribosomal protein mutants (*Shi et al., 2017*; *Xue and Barna, 2012*). In another scenario, in Diamond-Blackfan anemia patients carrying mutations in several ribosomal proteins, a lower ribosome content alters the translation of mRNAs critical for hematopoiesis (*Khajuria et al., 2018*). In yeast, lower ribosome levels may lead to dose-dependent changes in gene expression that parallel the overall growth rate (*Cheng et al., 2019*). A kinetic model of translation proposed that specificity could arise from disproportional effects on translational efficiency of specific mRNAs when the ribosome content in the cell changes (*Lodish, 1974*; *Mills and Green, 2017*). Overall, to understand how ribosomal protein mutants have distinct phenotypes, it is necessary to identify the relevant mRNAs with altered translational control, and then link the corresponding gene products to the observed phenotypes.

In budding yeast, 15 out of a total of 79 cytoplasmic ribosomal proteins are not essential (*Steffen et al., 2012*). Pairs of similar paralogs encode 59 of the ribosomal proteins (*Wapinski et al., 2010*). In most cases, cells lacking one of the two paralogs encoding a ribosomal protein are viable (*Giaever et al., 2002*; *Steffen et al., 2012*). Interfering with ribosome biogenesis usually affects cell cycle progression dramatically, but not uniformly across different ribosome biogenesis mutants (*He et al., 2014*; *Hoose et al., 2012*; *Komili et al., 2007*; *Thapa et al., 2013*). For example, loss of Rpl22Bp does not lengthen the G1 phase of the cell cycle, while the loss of Rpl22Ap does (*He et al., 2014*; *Hoose et al., 2012*; *Truong et al., 2013*). Dysregulation of translation is also strongly linked with aging. The number of times a yeast cell can divide and generate daughter cells defines its replicative lifespan (*Steffen et al., 2009*). Protein synthesis is dysregulated in aged cells, and it is thought to be a driver of aging (*Janssens et al., 2015*). Mutations in ribosomal proteins of the large (60S) subunit constitute a significant class of pro-longevity mutations in yeast and other species (*Kaeberlein and Kennedy, 2011*; *Kaeberlein et al., 2005*; *McCormick et al., 2015*; *Steffen et al., 2008*; *Steffen et al., 2012*). The *rpl* association with longevity, however, is often paralog-specific and complex. For example, the Rpl22 double paralog deletion is viable, but not long-lived (*Steffen et al., 2012*). The single *rpl22aΔ* mutants is long-lived, but *rpl22bΔ* cells are not long-lived (*Steffen et al., 2012*). In other ribosomal proteins, e.g., Rpl34, loss of either of the Rpl34 paralogs promotes longevity (*Steffen et al., 2012*). Importantly, bulk inhibition of translation with cycloheximide at various doses does *not* increase lifespan (*Steffen et al., 2008*). The above observations argue that simple relations between ribosome content, protein synthesis capacity, or generation time cannot sufficiently explain the longevity of *rpl* paralog mutants. To account for these paralog-specific phenotypes, we decided to identify patterns of translational control that are specific to paralogous ribosomal proteins and responsible for the increased longevity and altered cell cycle progression of *rpl* mutants.

Here, we identified changes in gene expression and metabolite levels that explain the differential longevity of Rpl22 paralog mutants. We show that translational control of enzymes involved in one-carbon metabolic pathways underpins replicative lifespan. Loss-of-function mutations in enzymes of these metabolic pathways extended the lifespan of otherwise wild type cells, underscoring the physiological relevance of our findings. Given the broad conservation of these pathways in other organisms, including humans, our results could have significant implications for longevity interventions.

## Results

### Rationale and experimental overview

Based on recent elegant studies (*Cheng et al., 2019*; *Khajuria et al., 2018*), lower ribosome levels and the accompanying longer generation times could underlie some of the phenotypes of ribosomal protein mutants. Hence, we first examined if generation time is associated with the replicative lifespan of *rpl* mutants. A weak, positive association had been reported between the change in mean lifespan in *rpl* mutants and their generation time relative to wild type cells (*Steffen et al., 2008*). Because ribosomal protein mutants often accumulate suppressors, we re-examined the association between lifespan and generation time using data from the fresh, recreated collection of all of the single ribosomal protein deletions (*McCormick et al., 2015*; *Steffen et al., 2012*). We also examined the relationship between lifespan and ribosomal protein abundance, using the latest consensus estimates of protein abundances in yeast (*Ho et al., 2018*). We found no significant association between the lifespan of *rpl* mutants with either their generation time ($\rho = -0.02$, based on the nonparametric, Spearman rank correlation coefficient), or the levels of the corresponding Rpl protein in wild type cells ($\rho = -0.06$; *Figure 1* and *Figure 1—source data 1*). Therefore, the general effects on generation time from ribosomal protein loss cannot adequately explain the longevity phenotypes of *rpl* mutants.

### Ribosome composition is largely unaffected in *rpl22* mutants

To identify the molecular basis for the paralog-specific *rpl* phenotypes, we focused primarily on the Rpl22Ap and Rpl22Bp pair. We chose Rpl22 because there are phenotypes of interest seen in *rpl22aΔ* but not in *rpl22bΔ* cells: First, *rpl22aΔ* cells have a consistently longer replicative lifespan ($\approx 38\%$) than wild type, or *rpl22bΔ*, cells (*Steffen et al., 2012*). Second, *rpl22aΔ* cells have a longer

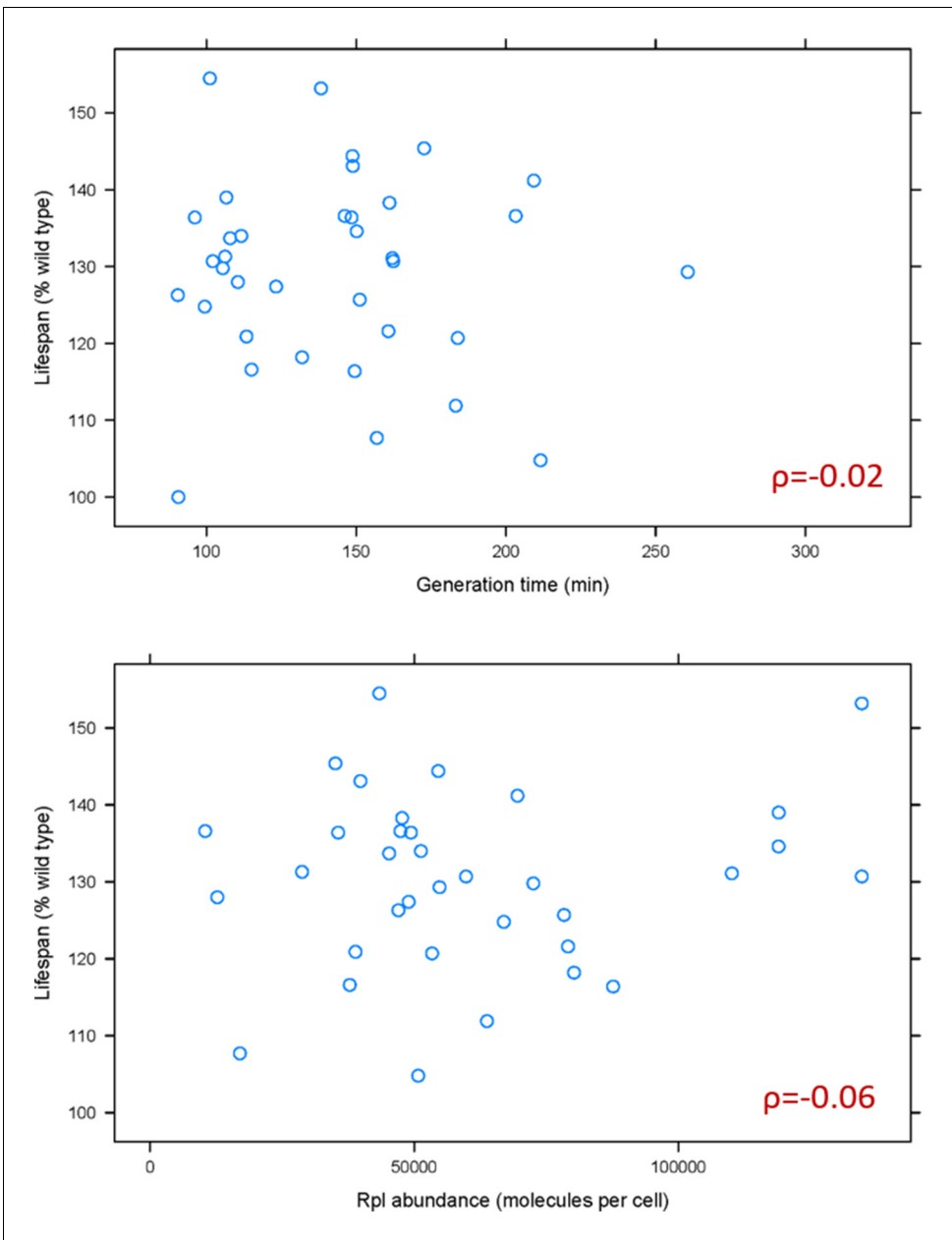

**Figure 1.** Doubling time and normal Rpl levels are not associated with the replicative lifespan of single *rpl* deletion mutants. (**A**) Scatterplot between the generation time (x-axis; from Tables S2 in *Steffen et al., 2012*) and replicative lifespan (y-axis; as percentage of the wild type lifespan, from Table 3 in *Steffen et al., 2012* and Table S2 in *McCormick et al., 2015*). (**B**) Scatterplot between the abundance of each deleted Rpl typically found in wild type cells (x-axis; the median number of molecules per cell, from Table S4 from *Ho et al., 2018*) and replicative lifespan (y-axis; as percentage of the wild type lifespan, from Table 3 in *Steffen et al., 2012* and Table S2 in *McCormick et al., 2015*). The Spearman correlation coefficients (ρ) shown in each case were calculated with the rcorr function of the *Hmisc* R language package, comparing the pair shown in each panel. All the values used as input for this figure and analyses are in *Figure 1—source data 1*.

The online version of this article includes the following source data for figure 1:

**Source data 1.** Lifespan, doubling time, and RP levels in ribosomal protein mutants.

G1 phase due to small birth size and a lower rate of size increase (*He et al., 2014*). Third, *rpl22a∆* cells are also more sensitive to oxidative stress (*Chan et al., 2012*).

We asked if the proportion of ribosomal proteins in assembled ribosomes changes significantly in *rpl22* deletion mutants. To this end, we isolated ribosomes through sucrose ultra-centrifugation from wild type, *rpl22a∆*, *rpl22b∆* or double *rpl22a,b∆* cells (*Steffen et al., 2012*), which were otherwise in the same isogenic haploid background (*Figure 2A*; see Materials and methods). We queried each strain in three independent replicates. Ribosomal protein abundance was measured with SWATH-mass spectrometry (see Materials and methods). We confirmed that in wild type (WT) cells, the levels of Rpl22Ap are much higher (>10 fold; *Figure 2B*) than Rpl22Bp (see also *Ho et al., 2018*). In *rpl22a∆* cells, there is a compensatory (≈5-fold) increase in the levels of Rpl22Bp, but the total Rpl22 levels are still significantly lower (≈2 to 3-fold) than in wild type cells (*Figure 2B*). A similar compensatory change in Rpl22 paralog expression exists in mice (*O'Leary et al., 2013*).

Next, we looked at the abundance of all other ribosomal proteins we could identify with SWATH-mass spectrometry (*Figure 2C*). Other than Rpl22 itself, the only RPs with statistically significant

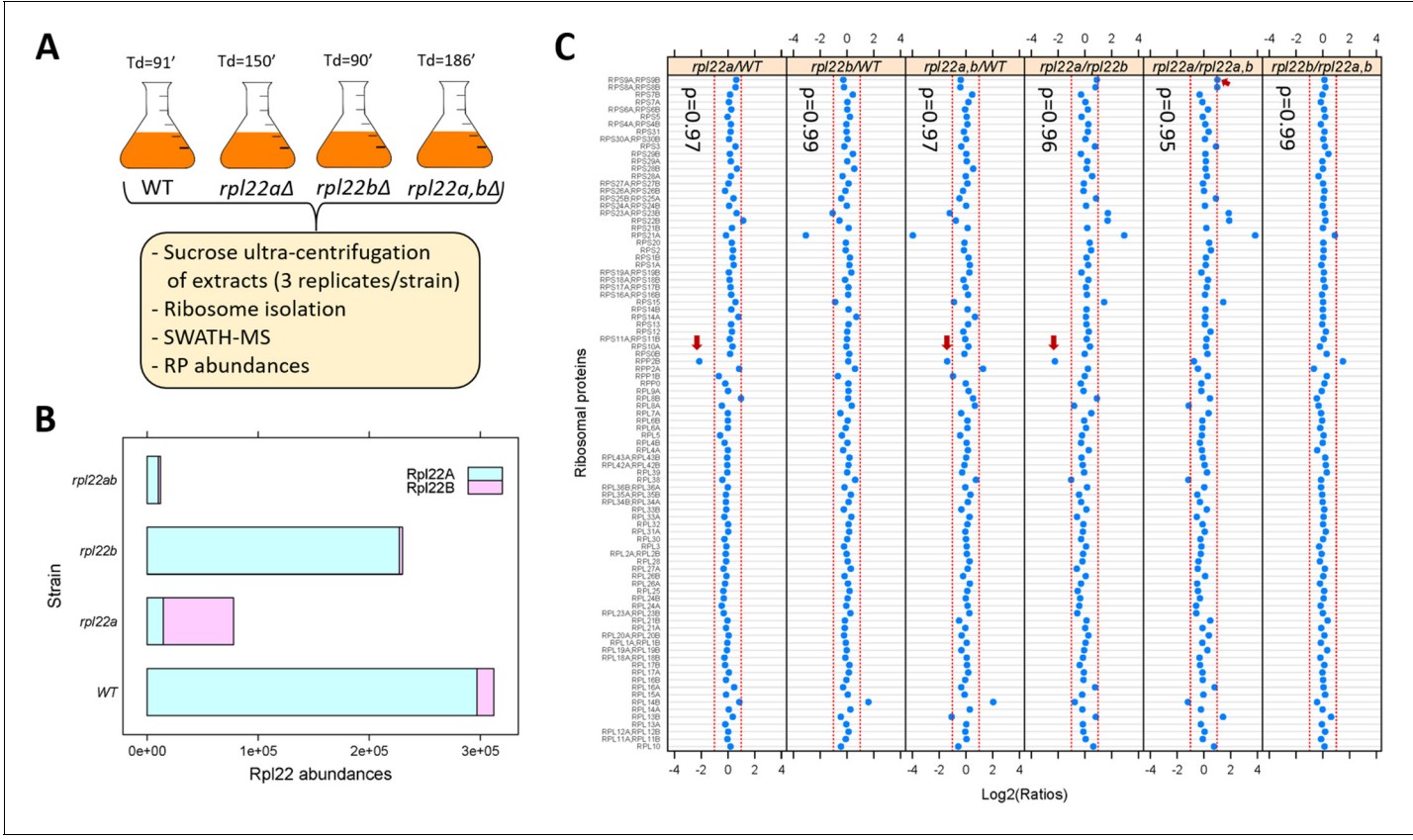

**Figure 2.** Loss of Rpl22 does not affect the relative abundance of other ribosomal proteins in ribosomes. (**A**) Schematic overview of the approach to query ribosomal protein abundances in *rpl22* deletion mutants (see Materials and methods). All strains were in the haploid BY4742 background. The doubling time values for each strain are shown on top (Td; in min, from Table S2 in *Steffen et al., 2012*). (**B**) Bar plot displaying on the x-axis the levels of Rpl22Ap and Rpl22Bp in wild type and *rpl22* deletion mutants, as indicated on the y-axis. The abundances shown correspond to the peak area intensities obtained from SWATH-MS (see Materials and methods and *Figure 2—source data 1*). (**C**) The Log2-transformed ratios of the intensities corresponding to the ribosomal proteins detected are indicated for each pairwise comparison among the 4 strains we analyzed. Changes higher than 2-fold are outside the dashed red lines in each panel. Note that not all the changes were statistically significant, as indicated. The red arrows indicate the only cases where the differences were significant (Log2FC ≥ 1 and p-value<0.05; bootstrap-based ANOVA; see Materials and methods) The Spearman correlation coefficients (ρ) shown in each case were calculated with the rcorr function of the *Hmisc* R language package, comparing the pair shown in each panel.

The online version of this article includes the following source data and figure supplement(s) for figure 2:

**Source data 1.** SWATH-mass spectrometry measurements of ribosomal protein abundances.

**Figure supplement 1.** Reduced protein synthesis in cells lacking Rpl22Ap.

differences higher than 2-fold among the four strains (indicated with the red arrows in *Figure 2C*) was Rpp2Bp (a stalk protein that does not interact with rRNA and may have been lost during ribosome isolation) and Rps9p (in only one binary comparison). In all strains tested, including the *rpl22a, b∆* cells lacking Rpl22 altogether, the relative proportion of the RPs in ribosomes was essentially constant, indicated by the Spearman rank correlation coefficients (ρ), which were very high (≥0.95) in each case. Hence, at least based on these population-averaged measurements, other than Rpl22 itself, the ribosomal composition seems unaffected by the loss of one or both of the Rpl22 paralogs.

## Loss of Rpl22Ap reduces overall protein synthesis

To quantify overall protein synthesis in wild type, *rpl22a∆*, *rpl22b∆* or double *rpl22a,b∆* cells, we measured incorporation of the methionine analog L-homo-propargylglycine (HPG) into newly synthesized proteins (*Figure 2—figure supplement 1A*). After the incorporated HPG was chemically modified to fluoresce (see Materials and methods), the fluorescence per cell was recorded by flow cytometry (*Figure 2—figure supplement 1B*). The cells were also imaged by microscopy (*Figure 2—figure supplement 1C*). Both *rpl22a∆* and *rpl22a,b∆* cells had similar and significantly reduced (≈2-fold) HPG incorporation compared to wild type or *rpl22b∆* cells (*Figure 2—figure supplement 1B*). Because *rpl22a,b∆* cells are not long-lived (*Steffen et al., 2012*), we conclude that merely reducing rates of protein synthesis is not sufficient to promote longevity. This conclusion is in agreement with the observation that inhibiting protein synthesis with cycloheximide also does not increase lifespan (*Steffen et al., 2008*). Hence, translational effects responsible for the longevity of *rpl22a∆* cells probably involve more nuanced and specific outputs, not reflected by bulk impacts on protein synthesis (*Figure 2—figure supplement 1B*) or cell generation time (*Figure 1* and *Figure 1—source data 1*).

## Generating RNAseq and Riboseq libraries from synchronous, dividing cells lacking ribosomal protein paralogs

Ribosomal protein mutants often have distinct cell cycle phenotypes, even when their generation time is similar (*He et al., 2014*). To capture translational effects in the different ribosomal protein paralog mutants that might be cell cycle-dependent, and difficult to discern from asynchronous cultures, we made our RNA libraries for ribosome profiling from highly synchronous cultures. We used centrifugal elutriation to obtain our synchronous cultures of ribosomal protein paralog mutants. Unlike arrest-and-release synchronization approaches, centrifugal elutriation maintains as much as possible the normal coupling of cell growth with cell division (*Aramayo and Polymenis, 2017*; *Soma et al., 2014*). To collect enough cells for these experiments, we followed the same approach as in our previous work on wild type cells (*Blank et al., 2017*). Briefly, for each mutant strain, elutriated G1 cells were allowed to progress in the cell cycle until they reached the desired cell size, at which point they were frozen away, and later pooled with cells of similar size (*Figure 3A*). In this manner, we collected enough cells to generate a cell size-series for each ribosomal protein mutant, spanning a range from 40 to 75 fL, sampled every 5 fL intervals in three biological replicates in each case (*Figure 3B*). Note that for these experiments, we used homozygous diploid strains lacking Rpl22Ap, Rpl22Bp, Rpl34Ap, or Rpl34Bp, for two reasons: First, to minimize the possible effects of recessive suppressors. Second, so that these datasets could be compared to a similar dataset we generated from the parental diploid wild type cells (*Blank et al., 2017*).

Budding coincides with the initiation of DNA replication and exit from the G1 phase in *S. cerevisiae* (*Guo et al., 2004*; *Park et al., 2013*; *Pringle and Hartwell, 1981*). Hence, we scored by microscopy the percentage of budded cells across each cell size series (*Figure 3B*), to gauge the synchrony we obtained. In every case, the pools of the smallest size (40 fL) were unbudded, rising steadily to >80% budded at the largest cell size (75 fL). The cell size at which half the cells are budded is often called the 'critical' size, serving as a convenient proxy for the commitment step START (*Hoose et al., 2012*). All four ribosomal protein mutants had a critical size of ≈60–65 fL (*Figure 3B*), which is similar to the critical size obtained from typical time-series elutriation experiments for wild type, *rpl22a∆/rpl22a∆*, or *rpl34a∆/rpl34a∆* cells we had reported previously (*He et al., 2014*). From the RNAseq data that we will describe later (*Figure 4*), transcripts that are known to increase in abundance in S (histone H4; *HHF1*), or G2 phase (cyclin *CLB2*), peaked as expected in the cell size series (*Figure 3C*). Overall, based on morphological (*Figure 3B*) and molecular (*Figure 3C*) markers

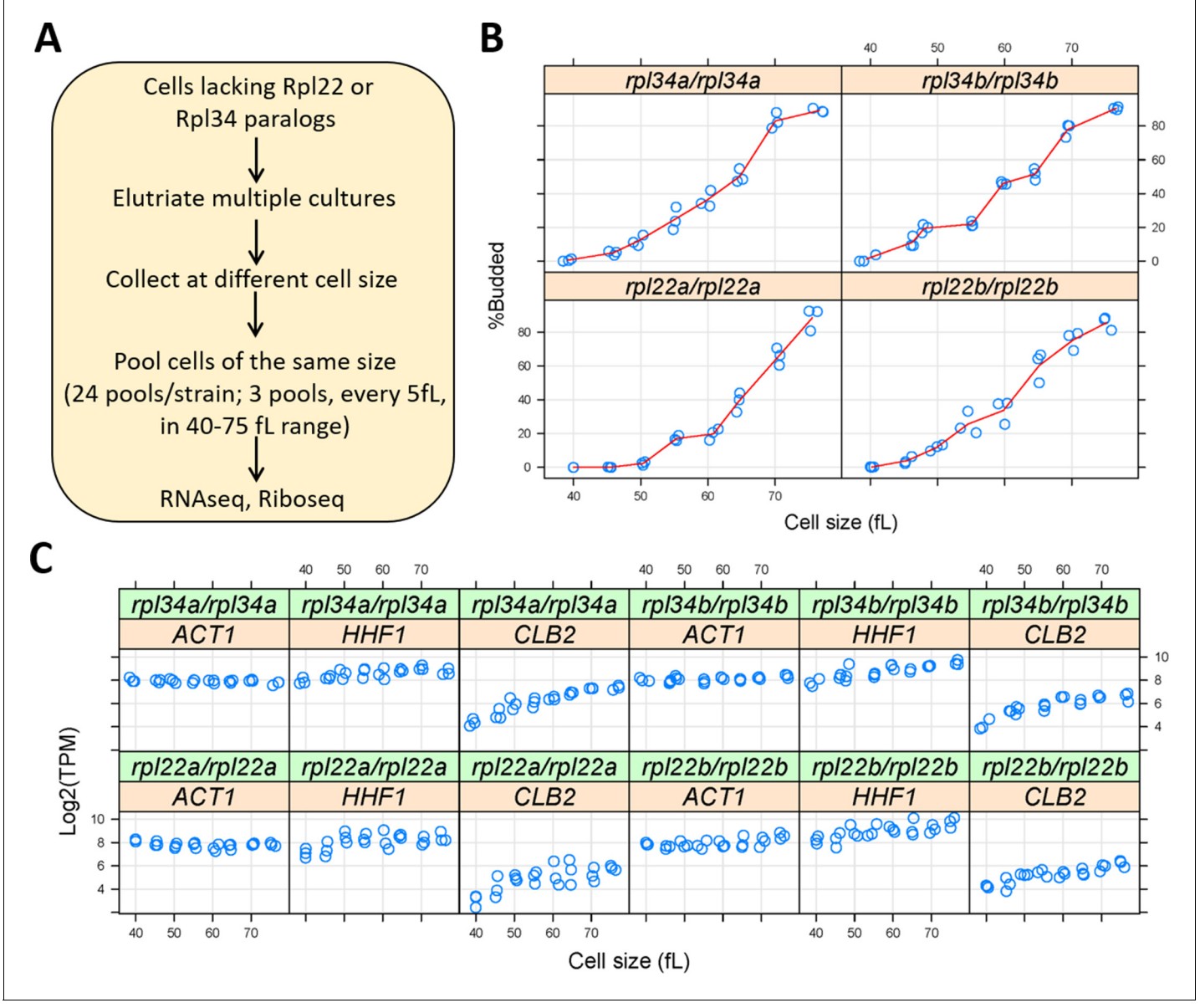

**Figure 3.** Querying synchronous, dividing cells lacking ribosomal protein paralogs. (**A**) Schematic of our experimental approach to identify gene expression changes in *rpl22* and *rpl34* paralog mutants at the transcriptional or translational level, from cells at several different stages of the cell cycle. (**B**) Cell size (in fL, x-axis) and the percentage of budded cells (%Budded, y-axis) from each pool of the indicated strains. The values shown are the weighted averages, from the different elutriated samples in each pool. (**C**) On the y-axis, are the Log2-transformed TPM values from representative transcripts known to increase in abundance in S (histone H4; *HHF1*), G2 phase (cyclin *CLB2*), or are constitutively expressed (*ACT1*). Cell size is on the x-axis.

The online version of this article includes the following source data for figure 3:

**Source data 1.** Raw read sequencing data.
**Source data 2.** TPM-normalized sequencing data.

of cell cycle progression, the synchrony of all the cell size series was of good quality. Note that our cell size series did not necessarily reflect the entire cell cycle. For example, *rpl22aΔ/rpl22aΔ* cells stay longer in the G1 phase because they are born smaller and also grow in size slower (*He et al., 2014*). Nonetheless, for any given cell size, the datasets we generated are directly comparable among the different strains, even though the cell cycle length of these strains may not be the same.

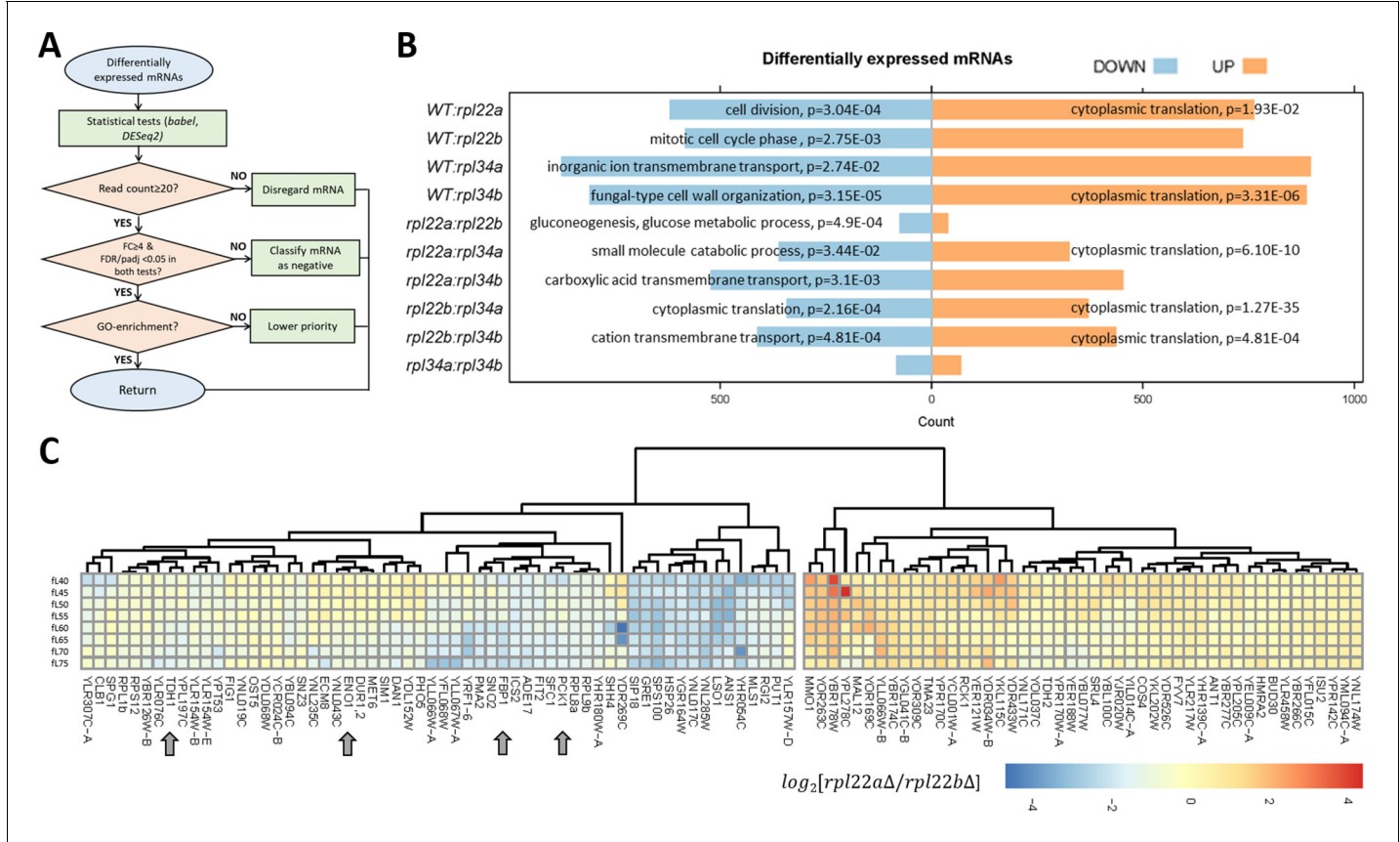

**Figure 4.** Transcripts with altered relative abundance in *rpl22* and *rpl34* paralog deletion mutants. (A) Decision diagram for identifying transcripts that were differentially expressed in paralog deletion mutants. (B) The number of transcripts with significantly different levels (adjusted p-value/FDR < 0.05, Log2FC $\geq$ 2, identified both from the *babel* and *DESeq2* R language packages) at any one cell size pool, between any pairwise comparison between WT, *rpl22*, and *rpl34* mutants is shown. The data for wild type diploid cells were from *Blank et al., 2017*. The Gene Ontology terms (*The Gene Ontology Consortium, 2019*) that were most enriched in each case are shown, based on the PANTHER (*Mi et al., 2019*) platform classification, incorporating the Holm-Bonferroni correction. All the gene names of loci with significant changes in transcript levels are in *Figure 4—source data 1* (sheet 'cumulative'). The 'UP' and 'DOWN' groupings correspond to each pairwise comparison shown, with the 'UP', or 'DOWN', group higher, or lower, for the strain in the numerator of the ratio, respectively. (C) Reduced levels of transcripts encoding glycolytic/gluconeogenic enzymes in *rpl22a*Δ compared to *rpl22b*Δ cells. The Log2-transformed ratio of the corresponding TPM values is shown in each case, across the different cell size pools. The data were hierarchically clustered without supervision and displayed with the *pheatmap* R package. Each row corresponds to a separate 5 fL cell size interval, from 40 to 75 fL, from top to bottom. The arrows indicate transcripts for key enzymes of glycolysis and gluconeogenesis. mRNAs with missing values across the cell sizes we analyzed were not included in the heatmap.

The online version of this article includes the following source data for figure 4:

**Source data 1.** mRNAs with significantly different abundances between any two strains.

From the samples described above, we prepared 96 ribosome footprint libraries (24 per ribosomal protein mutant) and the corresponding, sample-matched 96 RNA libraries. The libraries were constructed as described in our previous work for the wild type cells (*Blank et al., 2017*), with one significant exception. Here, we used rRNA subtraction for the RNA libraries, because it has since been reported that polyA-selected RNA might introduce bias in the quantification of the transcript abundances (*Weinberg et al., 2016*). The sequences were mapped as we describe in Materials and Methods. Using the same mapping pipeline, we re-mapped our published reads from the wild type cells as well (*Blank et al., 2017*). Together, these datasets comprise 120 ribosome footprint libraries and the matching 120 transcript libraries, used as input in all our subsequent analyses described in this report (see *Figure 3—source data 1*).

## Transcripts with altered relative abundance in *rpl22* and *rpl34* mutants

To identify differentially expressed mRNAs, we examined all ten pairwise comparisons among the *rpl22* and *rpl34* deletion mutants and wild type cells. In our computational pipeline (see *Figure 4A*, and Materials and methods), only ORFs with ≥20 reads were included. We used two different, R language-based packages, *babel*, and *DESeq2*, in each analysis. Candidate mRNAs with differential expression between two strains, at any one point in the cell cycle, had an adjusted p-value or false discovery rate (FDR) of <0.05 in *both* analyses, and a fold-change ≥4. Lastly, transcripts of the highest priority were those enriched in a gene ontology category (p<0.05, incorporating the Holm Bonferroni correction). Overall, at the transcript abundance level, no process was uniformly upregulated in the ribosomal protein mutants (*Figure 4B*, left). However, from all pairwise comparisons, two significant patterns emerge: First, the only process that was downregulated transcriptionally in *rpl* paralog mutants was cytoplasmic translation itself (*Figure 4B*, right). Second, when comparing paralog deletions of the same ribosomal protein (i.e., *rpl22aΔ* vs. *rpl22bΔ*; or *rpl34aΔ* vs. *rpl34bΔ*), there were few significant changes, compared to any other pairwise comparison. There was no gene ontology enrichment between the Rpl34 paralog deletions (*Figure 4B*). We noticed, however, that compared to *rpl22bΔ* cells, *rpl22aΔ* cells had significantly reduced (p=4.9E-04) levels of transcripts encoding enzymes of glycolysis and gluconeogenesis (*Figure 4C*): Tdh1p is glyceraldehyde-3-phosphate dehydrogenase; Eno1p is enolase; Fbp1 is fructose-1,6-bisphosphatase; Pck1p is phosphoenolpyruvate carboxykinase. These data suggest that compared to cells lacking Rpl22Bp, the central pathway of glycolysis/gluconeogenesis may be downregulated in cells lacking Rpl22Ap.

## Transcripts with altered relative translational efficiency in *rpl22* and *rpl34* mutants

To identify mRNAs with altered translational efficiency between the strains we queried, we relied again on multiple computational pipelines (*Figure 5—figure supplement 1A*, see Materials and Methods). Transcripts whose translational efficiency (i.e., the ratio of ribosome-bound to total mRNA abundance) was different between two strains, at any one point in the cell cycle, with an adjusted p-value or false discovery rate (FDR) of <0.05 in *all* three analyses, and a fold-change ≥2, were categorized as 'positive'. Our strategy of relying on multiple computational approaches to identify mRNAs under translational control in the various strains we analyzed may be very stringent in some cases, missing true positives. For example, *GCN4* was identified in several comparisons by one or more (see *Figure 5—source data 1*), but not all three of the computational pipelines we used. *GCN4* encodes a bZIP transcriptional activator of amino acid biosynthetic genes, whose expression responds to amino acid starvation. In wild type cells growing in nutrient-rich conditions, translation of *GCN4* is repressed by uORFs that impede ribosomes from initiating Gcn4p synthesis, while in poor nutrients, translation of *GCN4* is de-repressed (*Hinnebusch, 1985*). It is also known that the long-lived *rpl31aΔ* and *rpl20bΔ* ribosomal mutants have higher levels of Gcn4p (*Steffen et al., 2008*). Furthermore, Gcn4p is required for the full increase in longevity of *rpl* mutants (*Steffen et al., 2008*), and overexpression of Gcn4p can extend replicative lifespan without changes in bulk protein synthesis (*Hu et al., 2018*). As we mentioned above, in pairwise strain comparisons *GCN4* was not identified by all computational approaches we used (see *Figure 5—source data 1*). However, when examined in each strain against all other mRNAs of that strain, the translational efficiency of *GCN4* was significantly reduced in wild type and *rpl22bΔ* cells, but not in the long-lived *rpl22aΔ* cells (*Figure 5—figure supplement 2*), as expected from the results of *Steffen et al., 2008*. Overall, despite missing some target mRNAs, based on the data we will present next, the multiple computational pipelines of our approach yield robust, physiologically relevant datasets.

Based on ontology enrichment from the mRNAs we identified, we observed the following: First, compared to wild type cells, each *rpl* paralog mutant had upregulated the translational efficiency of mRNAs encoding gene products involved in cytoplasmic translation (*Figure 5—figure supplement 1B* and *Figure 5—figure supplement 3*). Cheng et al reported recently a similar phenomenon in *rps* mutants, but not in the *rpl* mutants they analyzed, which did not include *rpl22* or *rpl34* (*Cheng et al., 2019*). Hence, our results may reflect a more general response in *rp* mutants, including of the large 60S subunit, to compensate for their dysfunctional protein synthesis. Second, relative to wild type, the *rpl34aΔ*, and *rpl34bΔ*, paralog deletions had reduced translational efficiency of one-carbon (*Figure 5—figure supplement 3C*), and amino-acid (*Figure 5—figure supplement 3D*),

metabolic pathways, respectively. The down-regulation of mRNAs encoding 1C enzymes in *rpl34aΔ* cells is consistent with the hypothesis that down-regulation of 1C metabolism is associated with increased longevity, as we will describe later in the manuscript. Third, as was the case for the transcriptomic comparisons (*Figure 4B*), the smallest number of mRNAs with altered translational efficiency was observed when comparing paralog deletions of the same ribosomal protein (i.e., *rpl22aΔ* vs. *rpl22bΔ*; or *rpl34aΔ* vs. *rpl34bΔ*). For the *rpl34aΔ* vs. *rpl34bΔ* comparison, there was no enrichment for any particular group (*Figure 5—figure supplement 1B*), consistent with the fact that the *rpl34* paralog mutants have similar phenotypes, both being long-lived. Fourth, compared to *rpl22bΔ* cells, *rpl22aΔ* cells had significantly reduced (p=2.93E-02) translational efficiency of transcripts encoding enzymes of serine and methionine metabolism, which are part of what is collectively known as one-carbon metabolic pathways (*Figure 5B*). We will expand more on the latter result in subsequent sections.

Lastly, we looked at mRNA features of translationally controlled mRNAs. For each mRNA with altered translational efficiency, we examined the lengths of the ORF, its 5'-leader, and its 5'-UTR (*Lin and Li, 2012*). We also looked at the abundance of the corresponding protein (*Ho et al., 2018*). We grouped all the mRNAs with altered translational efficiency in any of the *rpl* deletions compared to wild type (*Figure 5—figure supplement 4*, top), and we also compared mRNAs with differential translation between *rpl22aΔ* vs. *rpl22bΔ* cells (*Figure 5—figure supplement 4*, bottom). Overall, there were statistically significant differences, but the magnitude of the differences was not particularly striking (*Figure 5—figure supplement 4*). One general feature seems to be that mRNAs that encode for relatively abundant proteins were among those that were translationally controlled (either up or downregulated) in the WT vs. *rpl* comparisons and also in *rpl22aΔ* vs. *rpl22bΔ* cells (in this case, for the downregulated group; *Figure 5—figure supplement 4*).

The differences between *rpl34aΔ* vs. *rpl34bΔ* both in the abundance of mRNAs (*Figure 4B*) and in their translational efficiency (*Figure 5—figure supplement 1B*) were few and without any significant enrichment in any gene ontology group. These observations are consistent with the similar phenotypes observed in *rpl34aΔ* and *rpl34bΔ* cells, which both have a longer replicative lifespan (*Steffen et al., 2012*), and with the notion that longevity is associated with altered translational efficiency of specific mRNAs. Hence, for the remainder of this study, we focused instead on the comparison between *rpl22aΔ* vs. *rpl22bΔ*, using the molecular information we generated to explain the different phenotypes of the *rpl22* paralog pair of deletion mutants.

## Translational control of methionine and serine metabolic pathways in *rpl22aΔ* cells

Among the transcripts with reduced translational efficiency in *rpl22aΔ* vs. *rpl22bΔ* cells (*Figure 5A*), the only significantly enriched (17.6-fold, p=2.93E-02) gene ontology group was the 'serine family amino acid metabolic process' (GO:0009069). The group included the mRNAs encoding Shm2p (cytosolic serine hydroxymethyltransferase), Met3p (sulfate adenylyltransferase), Met17p (homocysteine/cysteine synthase), Met5p (sulfite reductase, β subunit), Gcv2p (mitochondrial glycine dehydrogenase, decarboxylating). These enzymes are also part of methionine (Met3,5,17p) and folate (Shm2p, Gcv2p) metabolic pathways, collectively known as one-carbon (1C) metabolism. Additional mRNAs with reduced translational efficiency in *rpl22aΔ* vs. *rpl22bΔ* cells encoded enzymes of these pathways, shown schematically in *Figure 5B*. For example, Ade1p (N-succinyl-5-aminoimidazole-4-carboxamide ribotide synthetase) and Ade17p (containing 5-aminoimidazole-4-carboxamide ribonucleotide transformylase and inosine monophosphate cyclohydrolase activities), are required for 'de novo' purine nucleotide biosynthesis. We were intrigued by these findings because the loss of Met3p in wild type cells is known to extend replicative longevity (*McCormick et al., 2015*), a phenotype shared with loss of Rpl22Ap (*Steffen et al., 2008*; *Steffen et al., 2012*). To confirm that Met3p levels were significantly lower in *rpl22aΔ* vs. *rpl22bΔ* cells, we crossed a strain expressing TAP-tagged Met3p from its endogenous chromosomal location, with either *rpl22aΔ* or *rpl22bΔ* cells. We then evaluated the independent segregants of these crosses by immunoblots (*Figure 5C*). We found that cells lacking Rpl22Ap had ≈4–5-fold lower levels of Met3p-TAP compared to cells lacking Rpl22Bp (*Figure 5C*), consistent with our ribosome profiling data. Compared to wild type cells (*RPL22*[+] in *Figure 5C*), Met3p abundance was lower in *rpl22aΔ* cells, but higher in *rpl22bΔ* ones (*Figure 5C*).

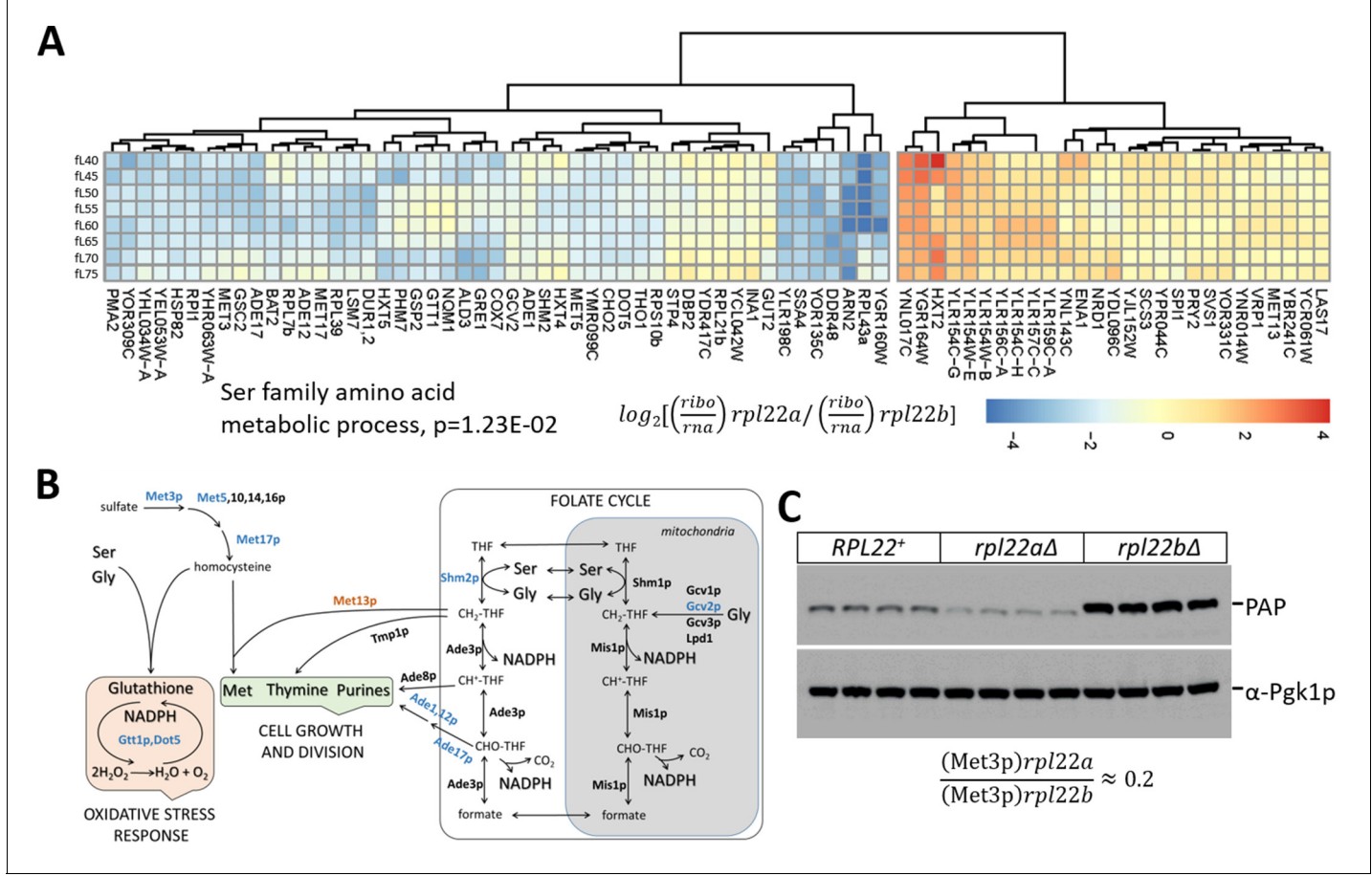

**Figure 5.** Reduced translational efficiency of transcripts encoding enzymes of the methionine and 1C metabolic pathways in *rpl22aΔ* cells. (**A**) Heatmap of the transcripts with significantly different translational efficiencies (see *Figure 3*) between the two Rpl22 paralog deletion strains. The ratio of the TPM values for the ribosome footprints ((ribo) against the corresponding values for the mRNA reads (rna) define the translational efficiency of each locus in each paralog mutant. The data were hierarchically clustered and displayed with the *pheatmap* R package. Each row corresponds to a separate cell size interval, from 40 to 75 fL, from top to bottom. The Gene Ontology terms highlighted were enriched, based on the PANTHER platform classification, incorporating the Holm-Bonferroni correction. (**B**) Diagram of the metabolic pathways involving enzymes whose mRNAs have altered translational efficiency in *rpl22aΔ* vs *rpl22bΔ* cells. Proteins whose mRNAs have lower translational efficiency in *rpl22aΔ* cells are shown in blue. (**C**) Immunoblots from WT (*RPL22⁺*), *rpl22aΔ*, or *rpl22bΔ* cells, carrying a *MET3-TAP* allele expressed from its endogenous chromosomal location. All the strains were otherwise isogenic. The signal corresponding to Met3p-TAP was detected with the PAP reagent, while the Pgk1p signal represents loading. The band intensities were quantified using the ImageJ software package, and the relative abundance of Met3p-TAP in *rpl22aΔ: rpl22bΔ* cells is shown at the bottom, from three independent such experiments.

The online version of this article includes the following source data and figure supplement(s) for figure 5:

**Source data 1.** mRNAs with significantly different translational efficiency between any two strains.
**Figure supplement 1.** Transcripts with altered translational efficiency (TE) in *rpl22* and *rpl34* paralog deletion mutants.
**Figure supplement 2.** The translational efficiency of *GCN4* is de-repressed in cells lacking Rpl22Ap.
**Figure supplement 3.** Dysregulation of the translational efficiency of transcripts encoding gene products involved in cytoplasmic translation in cells lacking Rpl22 or Rpl34 paralogs.
**Figure supplement 4.** Features of mRNAs with altered translational efficiency in *rpl* mutants.

Overall, only 52 mRNAs had reduced translational efficiency in *rpl22aΔ* vs. *rpl22bΔ* cells (*Figure 5A*). While additional targets may also contribute to the *rpl22aΔ* phenotypes, the one-carbon pathways involved could explain the phenotypic differences between the two Rpl22 paralog deletion mutants (*Figure 5B*). Note also that compared to wild type cells, in *rpl34aΔ* cells, the translational efficiency of enzymes of one-carbon metabolism was downregulated (*Figure 5—figure supplement 1B*). In addition to longevity effects attributed to Met3p, reduced expression of enzymes in one-carbon metabolic pathways could account for the longer G1 and slower growth of *rpl22aΔ* cells

(*He et al., 2014*). The same pathways are also responsible for the synthesis of glutathione (*Figure 5B*), potentially accounting for the sensitivity of rpl22aΔ cells to oxidative stress (*Chan et al., 2012*). As we described above, there were reduced levels of transcripts encoding key enzymes of glycolysis and gluconeogenesis in rpl22aΔ vs. rpl22bΔ cells (*Figure 4C*). Since serine biosynthesis relies on glycolytic intermediates, such as 3-phosphoglycerate (*Albers et al., 2003*), our transcriptomic and ribosome profiling data argue for the overall downregulation of 'serine family amino acid metabolic process' in rpl22aΔ cells. Additional metabolomic data we will present later support this conclusion.

## Metabolic profiling of *rpl22* paralog deletion mutants

Our transcriptomic and ribosome profiling datasets suggest reduced expression of enzymes of methionine and serine metabolic pathways in rpl22aΔ vs. rpl22bΔ cells. If so, then these changes in gene expression may be accompanied by changes in the levels of metabolites that are part of these pathways. To gain a better view of cellular physiology, we measured the levels of primary metabolites and biogenic amines in wild type, rpl22aΔ, and rpl22bΔ cells, through untargeted, mass spectrometry-based approaches (see Materials and methods). To pinpoint significant metabolic changes, we focused on the metabolites whose levels changed ≥2 fold between any two strains ($p<0.05$; based on bootstrapped ANOVA; see Materials and methods), and they were significantly enriched (FDR < 0.05) for a metabolic pathway (based on the MetaboAnalyst platform *Chong et al., 2019*). Cells lacking Rpl22Ap have lower levels of metabolites associated with central carbon (glycolysis, gluconeogenesis, pentose phosphate pathway), and amino acid (including glycine, serine, and methionine) metabolic pathways (*Figure 6A,B*). Exogenous addition of metabolites associated with the 1C pathways affected in rpl22aΔ cells did not suppress the slower growth of these cells on solid media (*Figure 6—figure supplement 1*). This was not unexpected, given the downregulation of central pathways such as glycolysis and gluconeogenesis (*Figures 4* and *6*). Lastly, rpl22aΔ cells had significantly elevated levels of the secondary metabolite nobiletin (*Figure 6—figure supplement 2*). Nobiletin is a flavonoid made by plants (*Ben-Aziz, 1967*), and it was presumably present in the rich, undefined medium we used in these experiments, probably in the yeast extract prepared at the end of industrial fermentations. Nobiletin has well-documented antioxidant properties (*Umeno et al., 2016*), which might explain why rpl22aΔ cells accumulate it, given that they are sensitive to oxidative stress (*Chan et al., 2012*). Overall, the metabolite data are in remarkable congruence with our transcriptomic (lower mRNA levels for enzymes in glycolysis and gluconeogenesis; *Figure 4*) and ribosome profiling (lower translational efficiency of mRNAs for enzymes in methionine and serine metabolic pathways; *Figure 5*) datasets.

We followed-up our untargeted metabolite measurements with a targeted, orthogonal approach that measures amino acid levels. We used the highly sensitive, HPLC-based, PTH amino acid analysis (*Heinrikson and Meredith, 1984*; see Materials and methods). We found that the Gly:Ser ratio and Trp levels are significantly lower in rpl22aΔ vs. rpl22bΔ cells (*Figure 6C*). The low Trp level in rpl22aΔ cells is also of interest because, as we have shown previously, lowering Trp levels promotes longevity in wild type cells (*He et al., 2014*).

## Genetic interventions in 1C metabolism that extend longevity

Since the expression of enzymes of one-carbon metabolism and the levels of associated metabolites were reduced in the long-lived rpl22aΔ cells (*Figures 5* and *6*), we asked if, besides *MET3* (*McCormick et al., 2015*), loss of other genes in the pathway may prolong replicative longevity. *ADE17* and *SHM2* encode 1C enzymes and they were translationally down-regulated in rpl22aΔ cells. *ADE2* does not encode a 1C enzyme and its translational efficiency did not change in rpl22aΔ vs. rpl22bΔ cells. We also tested shm1Δ cells, because even though the translational efficiency of *SHM1* was similar in rpl22aΔ vs. rpl22bΔ cells, Shm1p catalyzes in mitochondria the same reaction that Shm2p does in the cytoplasm. Loss of *ADE2* (encoding phosphoribosylaminoimidazole carboxylase) did not affect replicative lifespan (*Figure 7D*). On the other hand, loss of *SHM2*, *ADE17*, or *SHM1* (encoding the mitochondrial serine hydroxymethyltransferase) significantly extended the lifespan of otherwise wild type cells (*Figure 7*). We note that loss of Met3p leads to a similar magnitude (≈15%) of lifespan extension (*McCormick et al., 2015*). Interestingly, all three enzymes encoded by these genes, including Ade17p with its 5-aminoimidazole-4-carboxamide ribonucleotide

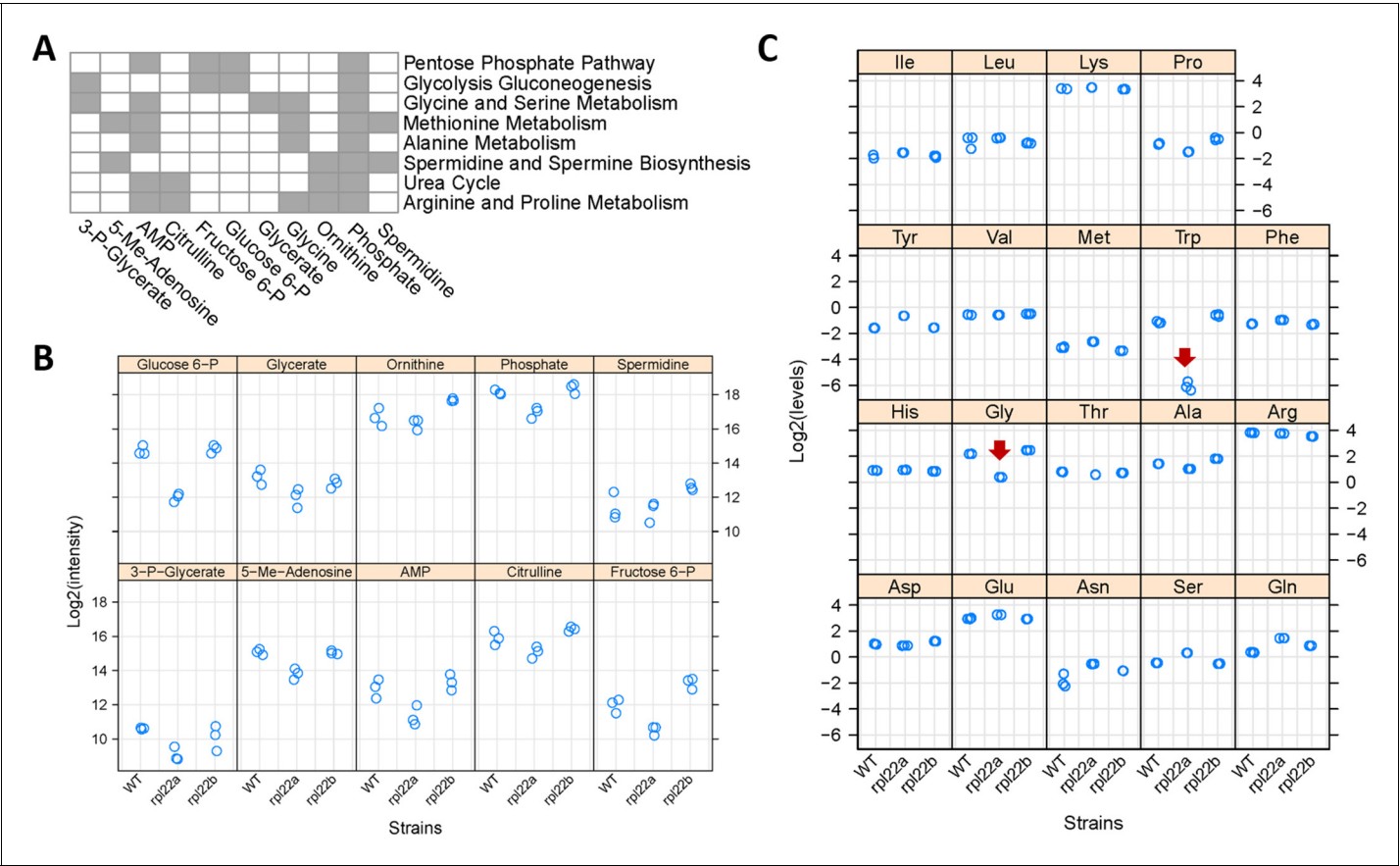

**Figure 6.** Metabolic profiling indicates reduced flux through central metabolic pathways and the folate cycle in *rpl22aΔ* cells. (**A**) Eleven metabolites shown at the bottom had significantly reduced levels in *rpl22aΔ* cells (Log2FC ≥ 1, p<0.05; based on bootstrapped ANOVA; see Materials and methods) and they were significantly enriched for the metabolic pathways shown to the right (FDR < 0.05). Pathway enrichment analysis was done with the *MetaboAnalyst* R language package. The metabolites were identified with untargeted, MS-based profiling of primary metabolites and biogenic amines, and targeted amino acid analysis. Metabolites indicated with gray in the Table are part of the pathways shown to the right. (**B**) The Log2-transformed peak intensities from the MS-based profiling of the metabolites shown in A (except Glycine) are on the y-axis. The strains used in the analysis are on the x-axis. (**C**) The Log2-transformed levels (in nmoles) of amino acids, after PTH-derivatization, Edman degradation and HPLC detection, are shown on the y-axis. The red arrows indicate the only amino acids (Gly and Trp) whose levels were significantly lower in *rpl22aΔ* cells (Log2FC ≥ 1, p<0.05; based on bootstrapped ANOVA; see Materials and Methods). The strains used in the analysis are on the x-axis, and they were in the BY4742 background.

The online version of this article includes the following source data and figure supplement(s) for figure 6:

**Source data 1.** Metabolite and amino acid abundances.
**Figure supplement 1.** Exogenous addition of metabolites from pathways affected in *rpl22aΔ* cells does not suppress the slower growth of these cells.
**Figure supplement 2.** Nobiletin is accumulated in *rpl22aΔ* cells.

transformylase activity, catalyze folate-dependent reactions (*Tibbetts and Appling, 2000*). In contrast, the carboxylase activity of Ade2p is not folate-dependent. These data suggest that genetic interventions in one-carbon metabolic pathways modulate longevity and provide strong support for the physiological relevance of our extensive profiling of long-lived *rpl* mutants.

To test the extent to which single deletions of 1C enzymes could contribute to longevity independently of *rpl22aΔ* cells, we measured the longevity of *rpl22aΔ* cells in the context of *ade17Δ*, *shm1Δ*, or *shm2Δ* deletions (*Figure 7—figure supplement 1*). In every case, the mean life span of the double mutants was the same or even somewhat decreased compared to single *rpl22aΔ* mutants (*Figure 7—figure supplement 1*). The effects did not reach statistical significance (p>0.05 in the log-rank test). We conclude that loss of 1C enzymes does not further extend the longevity of *rpl22aΔ* cells.

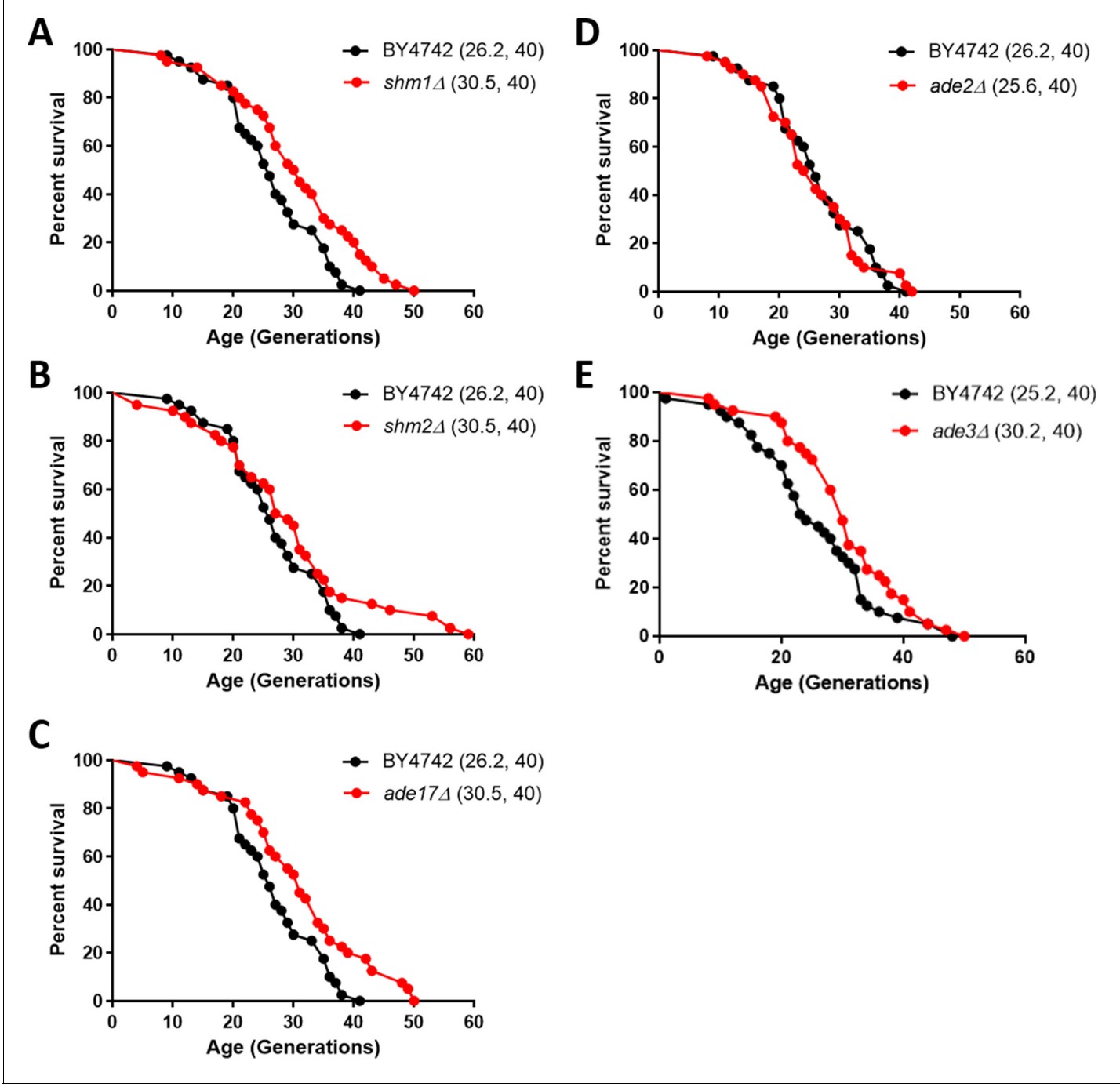

**Figure 7.** Deletion of enzymes of one-carbon metabolic pathways extends replicative lifespan in yeast. Survival curves for MATα (BY4742) cells (shown in black), compared to experiment-matched cells (shown in red) lacking *SHM1* (**A**), *SHM2* (**B**), *ADE17* (**C**), *ADE2* (**D**), or *ADE3* (**E**). Mean lifespans are shown in parentheses, along with the number of cells assayed in each case. In the case of *shm1Δ*, *shm2Δ*, *ade17Δ*, and *ade3Δ* cells, the lifespan extension was significant (p<0.0001; based on the log-rank test).

The online version of this article includes the following figure supplement(s) for figure 7:

**Figure supplement 1.** Deletion of enzymes of one-carbon metabolic enzymes does not further extend the replicative longevity of *rpl22aΔ* cells.

## Cell cycle control and 1C metabolic enzymes

We noticed that for several of the mRNAs with differential translation in *rpl22aΔ* vs. *rpl22bΔ* cells, the effect was cell cycle-dependent. For example, the translational efficiency of *MET3* and *ADE17* is disproportionately reduced early in the cell cycle in *rpl22aΔ* cells (*Figure 5A*). *SHM2* is an intriguing case because its translational efficiency appears to be highly cell cycle-dependent in wild type cells, peaking in the G1 phase (*Figure 8—figure supplement 1*). In our prior analysis of translational control in the cell cycle of wild type cells (*Blank et al., 2017*), we did not consider *SHM2* of high priority. Although both of the computational pipelines we had used at the time (*babel* and *anota*) identified *SHM2* as translationally controlled (see Dataset 8 in *Blank et al., 2017*), we had also applied a Fourier transform to identify the most periodic transcripts (*Blank et al., 2017*). Probably due to irregularities in the *SHM2* pattern (see *Figure 8—figure supplement 1*, leftmost panel), the Fourier-based method missed the periodic translational efficiency of *SHM2*. Note that for the data shown in *Figure 8—figure supplement 1*, the translational efficiency of *SHM2* is normalized against all the points in the cell cycle within each strain, not across the different strains. Strikingly, with the possible exception of *rpl22bΔ* cells, in all other *rpl* deletions, the translational efficiency of *SHM2* was not periodic, and there was not a peak in the G1 phase (*Figure 8—figure supplement 1*).

The roles of 1C pathways in cell division are prominent during DNA replication, but how they impact other phases of cell cycle progression is less clear. Furthermore, we had previously identified cell cycle alterations, specifically during G1 progression, which were associated with replicative longevity (*He et al., 2014*). Since we found that cells lacking 1C metabolic enzymes had increased lifespan (*Figure 7*), we next examined cell cycle progression when 1C metabolism was perturbed. We focused on cells lacking both Shm2p and Ade3p, which generate cytoplasmic 5,10-methylenetetrahydrofolate, because these cells were reported to be enlarged (*Yang and Meier, 2003*). Cell size changes may be indicative of cell cycle changes, and the enlargement of *shm2Δ, ade3Δ* cells is consistent with a DNA replication delay, analogous to the megaloblastosis observed in folate deficiencies (*Das et al., 2005*). We generated double mutant *shm2Δ, ade3Δ* cells, which were viable, but slow-growing (*Figure 8A*). With regards to cell size, we confirmed that *shm2Δ, ade3Δ* cells have a larger mean cell size (*Figure 8B*, left), but they are born at normal size (*Figure 8B*, right). However, the DNA content of *shm2Δ, ade3Δ* mutants was indistinguishable from that of wild type cells (*Figure 8C*). Hence, despite their enlargement and slower proliferation, on average, the relative duration of G1 and non-G1 cell cycle phases are not disproportionally affected in *shm2Δ, ade3Δ* cells, arguing for delays at multiple cell cycle phases. To determine whether the larger size of *shm2Δ, ade3Δ* cells resulted from a delayed G1/S transition, resembling mutants lacking G1 cyclins (*Blank et al., 2018*; *Cross, 1988*), or from a delay later in the cell cycle, we examined synchronous cultures obtained by elutriation. The rate *shm2Δ, ade3Δ* cells increase in size is about half that of wild type cells (*Figure 8D*), and the size at which they initiate DNA replication (a.k.a. critical size) is smaller (*Figure 8E*). Note that a reduction in the rate of cell size increase is associated with longer replicative longevity (*He et al., 2014*). Despite their smaller critical size, *shm2Δ, ade3Δ* cells have a substantially longer G1 (by ~170%) than wild type cells, because they reach that size much slower. We conclude that early in the cell cycle, cells lacking both Shm2p and Ade3p cannot increase in size fast enough, and their G1 is prolonged. After the G1/S transition, these cells are again delayed, presumably because of insufficient nucleotide pools for DNA replication. Still, during that time, they continue to increase in size, reaching a significantly larger size than that of wild type cells. These results show that perturbations of folate-based 1C pathways impact multiple cell cycle phases but in distinct ways. During G1 progression, the phenotypes of *shm2Δ, ade3Δ* cells are similar to other growth-limited mutants (e.g., ribosomal biogenesis mutants *He et al., 2014*). Later in the cell cycle, they resemble situations when nucleotide pools are depleted (e.g., upon exposure to hydroxyurea) when cells are delayed in the S phase (*Singer and Johnston, 1981*). Overall, our results point to the distinct impacts of 1C metabolism on cell cycle progression. 1C metabolism not only fulfills the metabolic needs for DNA replication but also supports cellular growth during G1 progression, perhaps accounting for the role of these pathways in cellular longevity.

## Discussion

In this report, we comprehensively examined paralog-specific, ribosomal protein mutants during cell division. The data we presented is significant for several reasons: First, we demonstrated that

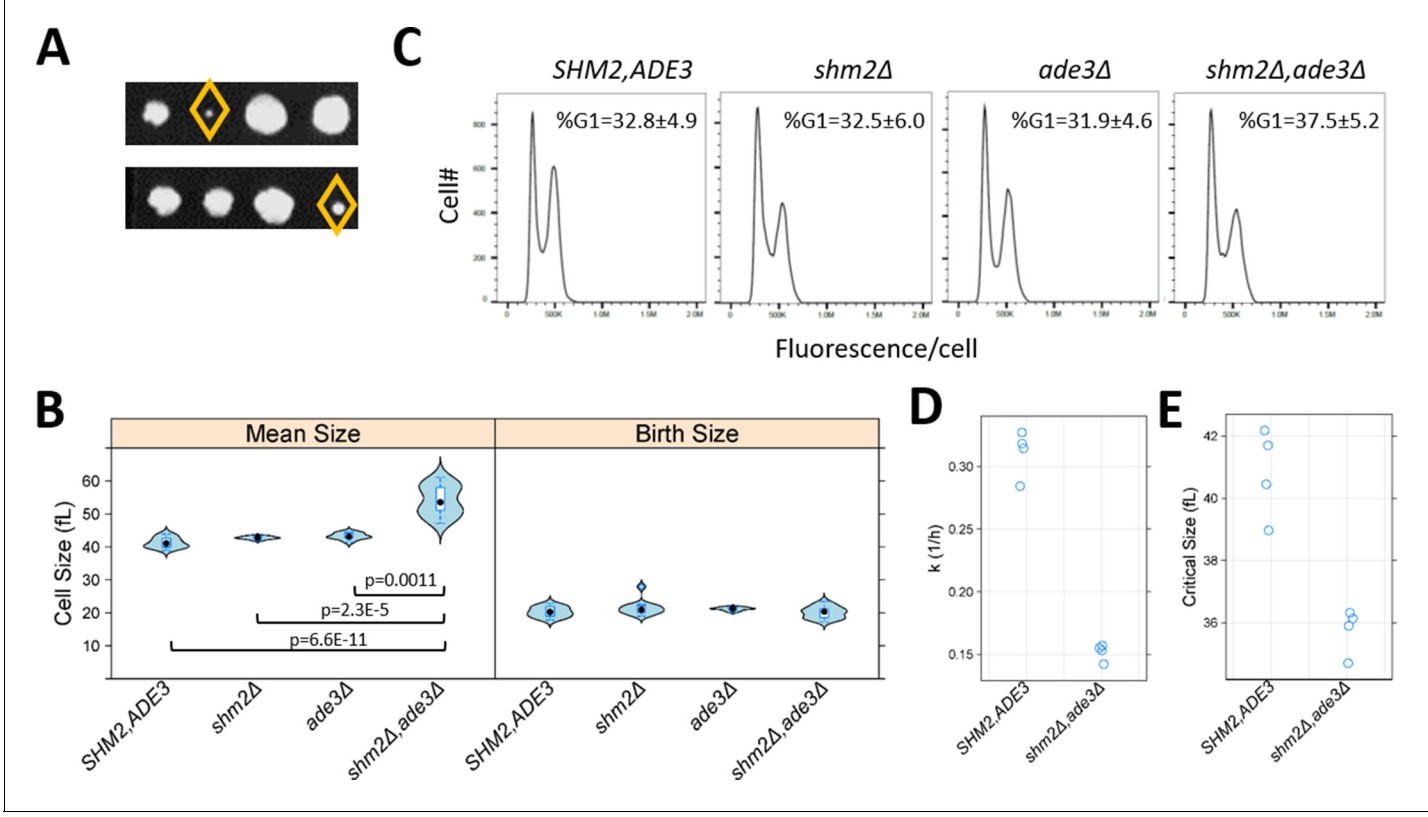

**Figure 8.** Loss of Shm2p and Ade3p impinges on multiple cell cycle phases but in distinct ways. (**A**) Double *shm2Δ,ade3Δ* deletion mutants are slow-growing. Two representative tetrad dissections from *shm2Δ* x *ade3Δ* crosses are shown. The yellow diamond indicates the *shm2Δ,ade3Δ* segregants. All the strains used in B-E were segregants from the same *shm2Δ* x *ade3Δ* crosses and, except as indicated, isogenic otherwise. (**B**) Violin plots of the mean and birth size of the indicated strains, calculated from ≥12 asynchronous cultures in each case. The plots were generated with the *lattice* R language package. Significant differences in pair-wise comparisons were indicated by the non-parametric Kruskal-Wallis test, while the p-values shown were for all significant differences based on the posthoc Nemenyi test (performed with the *PMCMR* R language package). (**C**) Representative DNA content histograms from the indicated strains, from at least 10,000 cells and ≥5 independent asynchronous cultures in each case. On the x-axis is fluorescence per cell, while the cell number is on the y-axis. The average and sd of the percentage of cells with unreplicated DNA (%G1) is shown. There were no statistically significant differences among the strains, based on the non-parametric Kruskal-Wallis test. (**D**) The rate of cell size increase (k, in h$^{-1}$, shown on the y-axis) was calculated as described in *Soma et al., 2014*, for the strains shown on the x-axis. Each data point is the average of two technical replicates, from synchronous elutriated cultures. (**E**) The critical size (y-axis) for the strains shown on the x-axis was calculated as described in *Soma et al., 2014*, from the same cultures as in D.

The online version of this article includes the following figure supplement(s) for figure 8:

**Figure supplement 1.** The translational efficiency of *SHM2* is cell cycle-regulated.

specific mRNAs are under translational control in those mutants. Second, the changes in gene expression we described, supported by metabolite measurements, explain the phenotypic differences of cells lacking Rpl22 paralogs. Third, the data underscore the role and physiological relevance of one-carbon metabolic pathways in cell division and longevity. Below, we discuss our results in the broader context of ribosomal protein perturbations and metabolic alterations in cell division and cellular aging.

## Accounting for specific phenotypes in ribosomal protein mutants

There is clear genetic evidence linking some ribosomal proteins to diseases (*De Keersmaecker et al., 2015*; *Mills and Green, 2017*). What is less clear, however, is how ribosomal protein mutations contribute to disease. Early in life, ribosomopathies are consistent with hypo-proliferation, such as defective hematopoiesis (*De Keersmaecker et al., 2015*). Perhaps paradoxically, later in life, some of these patients are predisposed to cancer (*Aspesi and Ellis, 2019*; *De Keersmaecker et al., 2015*). Rpl22, a ribosomal protein we focused on in this report for its association with longevity and

cell division in yeast, has also attracted a lot of attention in animal systems. For example, 10% of primary human samples of T-cell acute lymphoblastic leukemia have loss-of-function mutations in *RPL22* (*Rao et al., 2012*). *RPL22* is also mutated in microsatellite-unstable colorectal (*Ferreira et al., 2014*), and endometrial cancers (*Ferreira et al., 2014*; *Novetsky et al., 2013*), at 77%, and 50% frequency, respectively. In mice, the Rpl22 paralogs are *Rpl22* and *Rpl22-like1* (*Rpl22l1*). Mice lacking *Rpl22* are alive, but their αβ lineage of T cells does not develop (*Anderson et al., 2007*; *Stadanlick et al., 2011*). It turns out that in mice, the loss of *Rpl22* also triggers a compensatory increase in *Rpl22l1* expression (*O'Leary et al., 2013*), similar to what we saw in yeast (*Figure 2B*). Several models have been proposed to explain the phenotypes of ribosomal protein mutants. These models may not necessarily be exclusive of each other, and we discuss them briefly below.

Ribosomal proteins could have extra-ribosomal, non-translational functions (*Wang et al., 2015*; *Warner and McIntosh, 2009*). Free ribosomal proteins accumulate when ribosome biogenesis is perturbed, causing nucleolar stress. It has been reported that both Rpl22 and Rpl22l1 have extra-ribosomal, antagonistic functions in the nucleus (*Zhang et al., 2017*). Loss of Rpl22 in mice activates the stress-induced NF-κB pathway, increasing expression of the stemness factor Lin28B, which then leads to cancer (*Rao et al., 2012*). When ribosome assembly is disrupted, some of the released ribosomal proteins could bind other targets. For example, Rpl5, Rpl11, and Rpl23 have been reported to impinge on the p53 pathway. These ribosomal proteins inhibit an ubiquitin ligase (Mdm2 in mice, Hdm2 in humans) that degrades the p53 protein (*Wang et al., 2015*). How these effects could lead to cancer is not clear, since the outcome of those extra-ribosomal roles (i.e., stabilization of p53) would be *hypo*-proliferative. In our studies, we did not see any evidence for, or against, extra-ribosomal roles, since we focused on translational outputs. However, we note that at least in the case of *rpl22* mutants in yeast, translational outputs were sufficient to explain phenotypes of *rpl22aΔ* cells (*Figure 5*).

How could specific translational effects come about, from perturbations of an abundant and seemingly homogeneous cellular machinery as the ribosome? Impairing ribosomal proteins could alter the composition of active ribosomes (*Briggs and Dinman, 2017*; *Dinman, 2016*). Translation of some mRNAs may depend on such specialized ribosomes with different composition, accounting for the distinct phenotypes of ribosomal protein mutants (*Shi et al., 2017*; *Xue and Barna, 2012*). However, the extent to which ribosome specialization could, or is even necessary to, explain the phenotypes of ribosomal protein mutants is unclear (*Mills and Green, 2017*). Our data showing that ribosome composition was not significantly affected in *rpl22* mutants (*Figure 2C*) does not support the specialized ribosome hypothesis. Nevertheless, our data also does not necessarily refute the existence of specialized ribosomes because we performed population-averaged measurements. Translational effects specific to some mRNAs could also arise from the non-linear relationship between translational efficiency and the available ribosome content (*Lodish, 1974*). Ribosomal protein perturbations reduce the concentration of active ribosomes (*Cheng et al., 2019*; *Khajuria et al., 2018*; *Steffen et al., 2008*), which could then disproportionately affect translation of specific transcripts with elements that impede ribosome access to the main start codon of those mRNAs (*Khajuria et al., 2018*; *Mills and Green, 2017*).

To identify translational outputs from ribosomal protein perturbations, regardless whether they arise from altered ribosome composition or concentration, the translational effects on each mRNA need to be measured in ribosomal protein mutants, using unbiased, sensitive, genome-wide, ribosome profiling methods (*Ingolia et al., 2009*; *McGlincy and Ingolia, 2017*). To date, two prior studies queried ribosomal protein mutants with ribosome profiling methodology, in human cells (*Khajuria et al., 2018*) and yeast (*Cheng et al., 2019*). In both of these studies, specific translational outputs were detected, broadly related to altered ribosome levels, with no evidence for modified ribosome composition. In the yeast study (*Cheng et al., 2019*), there were also patterns of gene expression changes that reflected the doubling time of the ribosomal protein mutants. Overall, altered translational control as a function of ribosome content and, consequently, the growth rate is reasonable (*Lodish, 1974*; *Mills and Green, 2017*). In our previous work, we have also invoked these relationships to explain translational control of mRNAs encoding a G1 cyclin (*Blank et al., 2018*; *Polymenis and Schmidt, 1997*) and a lipogenic enzyme (*Blank et al., 2017*). However, as we will discuss next, growth rate-associated changes cannot explain all the *rp* mutant phenotypes. In yeast, it has also been recently questioned whether the growth rate is tightly associated with the ribosome content (*Kafri et al., 2016*; *Metzl-Raz et al., 2017*).

Since the longevity of *rpl* mutants is not significantly associated with generation time (*Figure 1*, see also *Steffen et al., 2008*), such simple relationships cannot explain the different behavior of *rpl* mutants with regards to longevity. It is also important to note that even when their overall generation time is similarly prolonged, ribosomal protein mutants may be delayed in the cell cycle for different reasons. We have shown that different variables of G1 progression (birth size, rate of size increase, or critical size) are differentially affected in *rp* mutants (*He et al., 2014*; *Soma et al., 2014*). Hence, using generation time/growth rate as an aggregate metric to classify the behavior of ribosomal protein mutants may mask the different causes that underlie their longer generation time and possibly other phenotypes these mutants may display. A unique aspect of our experiments is that we measured changes in translational efficiency at multiple points during synchronous, unperturbed cell cycle progression (*Figure 3*). This approach enabled us to probe at a much higher dynamic range translational outputs and capture cell cycle-dependent changes in translational efficiency. To our knowledge, there is no experimental support for altered levels of ribosomal proteins or ribosomes in the cell cycle (*Blank et al., 2020*; *Elliott et al., 1979*; *Polymenis and Aramayo, 2015*; *Shulman et al., 1973*). Therefore, the evidence for the cell cycle-dependent patterns of translational control we described here (*Figure 5*, *Figure 8—figure supplement 1*) and elsewhere (*Blank et al., 2017*) is consistent with added layers of regulation, distinct from growth rate-dependent changes in ribosome content.

## One carbon metabolism in cell division and cellular aging

The pathways of methionine, serine, glycine, and folates, collectively called one-carbon metabolism, have attracted enormous attention due to their involvement in diseases, especially cancer (*Ducker and Rabinowitz, 2017*; *Fox and Stover, 2008*; *Locasale, 2013*; *Newman and Maddocks, 2017*; *Rosenzweig et al., 2018*). The interest arises from the self-evident role of these folate-mediated transformations (see *Figure 5B*) in the biosynthesis of nucleotides (purines and thymidine; making the pathway the original (*Farber and Diamond, 1948*) and still widely-used chemotherapy target), amino acid homeostasis, epigenetic maintenance (through the methylation units the pathway provides), and redox defense (through the generation of glutathione). The folate cycle has parallel mitochondrial and cytoplasmic reactions (*Figure 5B*). One-carbon units enter through serine, glycine, or formate, while the outputs include thymidine, serine, methionine, purines, formate, carbon dioxide, and NAD(P)H (*Figure 5B*). The relative contributions of the mitochondrial pathways vs. the cytosolic ones vary, depending on nutrient and proliferation status, and the organism. In animal cells, most of the methionine 1C units originate from mitochondria (*Herbig et al., 2002*), while cytoplasmic serine hydroxymethyltransferase (cSHMT) directs 1C units toward dTMP synthesis (*Herbig et al., 2002*). Mutations in cSHMT are associated with diseases, including certain cancers (*Fox and Stover, 2008*).

In yeast, cells regulate the balance of one-carbon flow between the donors by controlling cytoplasmic serine hydroxymethyltransferase (Shm2p) activity (*Piper et al., 2000*). For example, by monitoring the levels of the 5,10-methylenetetrahydrofolate pool (usually derived from serine), cells can upregulate glycine catabolism for one-carbon generation when 5,10-methylenetetrahydrofolate is limiting or spare the breakdown of serine when glycine is in surplus (*Piper et al., 2000*). In cells lacking Rpl22Ap, we found that the translational efficiency of *SHM2* (*Figure 5*, *Figure 8—figure supplement 1*) and the Gly:Ser ratios (*Figure 6*) are low. These observations are consistent with a low flux through 1C pathways. We also documented a strong cell cycle-dependent control of the translational efficiency of *SHM2*, peaking in late G1 (*Figure 8*). Interestingly, an elevation of human cSHMT levels in S phase, specifically the SHMT1 isoform, has also been reported, without a change in *SHMT1* mRNA levels (*Anderson et al., 2012*; *Lan et al., 2018*).

We note that the cell cycle profile of wild type cells (33% of the cells in G1; *Figure 8C*) is not affected at all by the loss of Ade3p or Shm2p (32–33% G1 cells *Figure 8C*), and only slightly so by the loss of Shm1p or Ade17p (27–28% G1 cells; not shown). Likewise, the doubling time of single *ade3Δ*, *ade17Δ*, *shm1Δ* or *shm2Δ* cells is similar to the doubling time of wild type cells (*Giaever et al., 2002*; and not shown). However, the double *shm1Δ,shm2Δ* deletion is lethal (*Deutscher et al., 2006*), and the double *shm2Δ,ade3Δ* cells proliferate extremely slowly (*Figure 8A*). The redundancy of 1C pathways, e.g., with the near identical cytoplasmic and mitochondrial pathways (*Figure 5B*), likely accounts for these effects on cell cycle progression. The cell cycle phenotypes we described upon loss of Shm2p and Ade3p (*Figure 8*) illustrate the different

ways that folate-based 1C pathways can impinge on cell cycle progression. In other words, *shm2, ade3* mutants cannot *grow* fast enough in G1, but once they complete the G1 transition, they cannot *divide* fast enough. What is particularly interesting with the *shm2, ade3* mutants is not that DNA replication is delayed after it is started (leading to larger cell size), but that the cells also monitor folate-based transformations before they decide to initiate a new round of cell division, in the G1 phase. It is not known how the fluxes of the different outputs of folate metabolism change in the cell cycle. Our results argue for dynamic changes in these outputs, as cells transition through the different phases of the cell cycle. It is reasonable to speculate that folate-based reactions may be uniquely positioned to integrate growth cues and nutrient status with cell cycle progression.

Methionine restriction has been known to promote longevity in yeast and animals (*Johnson and Johnson, 2014*; *Lee et al., 2014*; *Lee et al., 2018*; *Ruckenstuhl et al., 2014*). By and large, however, 1C metabolism has not been previously implicated in longevity mechanisms. Consistent with 1C involvement in longevity pathways are recent observations linking the drug metformin with 1C metabolism (*Cuyàs et al., 2019*; *Menendez and Joven, 2012*). Metformin is known to promote longevity and is now in clinical trials for those effects (Clinical Trials Identifier: NCT02432287). In this context, our results that loss of core 1C enzymes (Shm1p, Shm2p, Ade17p) increase the mean and maximal lifespan of yeast cells (*Figure 7*) provide direct support for a role of folate-dependent metabolism in longevity mechanisms. It is likely that additional long-lived *rpl* mutants may have 1C metabolic alterations, similar to *rpl22aΔ*, since these mutants also display a slower growth rate, longer G1 phase (*He et al., 2014*), and reduced resistance to oxidative stress (*Ando et al., 2007*; *Brown et al., 2006*; *Okada et al., 2014*). With regards to longevity mechanisms, we note that it is only mild perturbations in 1C pathways that extend lifespan, in several contexts (e.g., in the long-lived *ade17Δ, shm1Δ, shm2Δ,* or *ade3Δ* cells; see *Figure 7*). It is also mild, but not severe delays in cell cycle progression that in some cases extend longevity (*He et al., 2014*).

In summary, our integrated transcriptomic, translatomic, and metabolomic datasets pinpoint mechanisms that control the expression of enzymes of methionine and serine metabolic pathways in the cell cycle and replicative longevity. These results explain the phenotypic differences of ribosomal protein paralog mutants and support a broad role for 1C pathways in cell division and cellular longevity.

## Materials and methods

**Key resources table**

| Reagent type (species) or resource | Designation | Source or reference | Identifiers | Additional information |
|---|---|---|---|---|
| Strain, strain background (*S. cerevisiae*) | BY4743 | *Giaever et al., 2002* | RRID:SCR_003093 | *MATa/α his3Δ1/his3Δ1 leu2Δ0/leu2Δ0 LYS2/ lys2Δ0 met15Δ0/MET15 ura3Δ0/ura3Δ0* |
| Strain, strain background (*S. cerevisiae*) | 32672 | *Giaever et al., 2002* | RRID:SCR_003093 | *rpl22aΔ::KanMX/rpl22aΔ:: KanMX,* BY4743 otherwise |
| Strain, strain background (*S. cerevisiae*) | 35844 | *Giaever et al., 2002* | RRID:SCR_003093 | *rpl22bΔ::KanMX/ rpl22bΔ::KanMX,* BY4743 otherwise |
| Strain, strain background (*S. cerevisiae*) | 30192 | *Giaever et al., 2002* | RRID:SCR_003093 | *rpl34aΔ::KanMX/ rpl34aΔ::KanMX,* BY4743 otherwise |
| Strain, strain background (*S. cerevisiae*) | 31445 | *Giaever et al., 2002* | RRID:SCR_003093 | *rpl34bΔ::KanMX/ rpl34bΔ::KanMX,* BY4743 otherwise |
| Strain, strain background (*S. cerevisiae*) | BY4742 | *Steffen et al., 2012* | | *MATα his3Δ1 leu2Δ0 lys2Δ0 ura3Δ0* |

*Continued on next page*

*Continued*

| Reagent type (species) or resource | Designation | Source or reference | Identifiers | Additional information |
|---|---|---|---|---|
| Strain, strain background (*S. cerevisiae*) | KS976 | *Steffen et al., 2012* | | *rpl22aΔ::URA3*, BY4742 otherwise |
| Strain, strain background (*S. cerevisiae*) | KS979 | *Steffen et al., 2012* | | *rpl22bΔ::URA3*, BY4742 otherwise |
| Strain, strain background (*S. cerevisiae*) | KS999 | *Steffen et al., 2012* | | *rpl22a,bΔ::URA3*, BY4742 otherwise |
| Strain, strain background (*S. cerevisiae*) | BY4741 | *Giaever et al., 2002* | RRID:SCR_003093 | *MATa his3Δ1 leu2Δ0 met15Δ0 ura3Δ0* |
| Strain, strain background (*S. cerevisiae*) | MET3-TAP | Dharmacon | YSC1178-202231887 | *MET3-TAP::HIS3M × 6*, BY4741 otherwise |
| Strain, strain background (*S. cerevisiae*) | HB147 | This study | | *rpl22aΔ::URA3*, *MET3-TAP::HIS3M × 6*, BY4741 otherwise |
| Strain, strain background (*S. cerevisiae*) | HB171 | This study | | *rpl22bΔ::URA3*, *MET3-TAP::HIS3M × 6*, BY4741 otherwise |
| Strain, strain background (*S. cerevisiae*) | 13403 | *Giaever et al., 2002* | RRID:SCR_003093 | *shm1Δ::KanMX*, BY4742 otherwise |
| Strain, strain background (*S. cerevisiae*) | 12669 | *Giaever et al., 2002* | RRID:SCR_003093 | *shm2Δ::KanMX*, BY4742 otherwise |
| Strain, strain background (*S. cerevisiae*) | 16561 | *Giaever et al., 2002* | RRID:SCR_003093 | *ade17Δ::KanMX*, BY4742 otherwise |
| Strain, strain background (*S. cerevisiae*) | 12384 | *Giaever et al., 2002* | RRID:SCR_003093 | *ade2Δ::KanMX*, BY4742 otherwise |
| Strain, strain background (*S. cerevisiae*) | 6591 | *Giaever et al., 2002* | RRID:SCR_003093 | *ade3Δ::KanMX*, BY4741 otherwise |
| Strain, strain background (*S. cerevisiae*) | NM64 | This study | | *MATα, shm2Δ::KanMX, ade3Δ::KanMX, met⁻, lys⁻* |
| Strain, strain background (*S. cerevisiae*) | NM65 | This study | | *MATa, shm2Δ::KanMX, ade3Δ::KanMX, met⁻, lys⁻* |
| Strain, strain background (*S. cerevisiae*) | NM66 | This study | | *MATα, ade17Δ::KanMX, rpl22aΔ::URA, his⁻, lys⁻, leu⁻* |
| Strain, strain background (*S. cerevisiae*) | NM67 | This study | | *MATα, shm2Δ::KanMX, rpl22aΔ::URA, his⁻, leu⁻, met⁻* |
| Strain, strain background (*S. cerevisiae*) | NM68 | This study | | *MATα, shm1Δ::KanMX, rpl22aΔ::URA, his⁻, leu⁻, met⁻* |
| Strain, strain background (*S. cerevisiae*) | BW885 | This study | | *MATα his3Δ1 leu2Δ0 lys2Δ0 ura3Δ0* |

*Continued on next page*

*Continued*

| Reagent type (species) or resource | Designation | Source or reference | Identifiers | Additional information |
|---|---|---|---|---|
| Other | Yeast extract | Sigma-Aldrich | Y1625 | |
| Other | Peptone | Sigma-Aldrich | P5905 | |
| Chemical compound, drug | Dextrose | Sigma-Aldrich | D9434 | |
| Chemical compound, drug | Cycloheximide | Calbiochem | 239763 M | |
| Chemical compound, drug | Sodium azide | Sigma-Aldrich | S2002 | |
| Chemical compound, drug | Tris (hydroxymethyl) aminomethane | Sigma-Aldrich | 252859 | |
| Chemical compound, drug | Tris base | Roche | TRIS-RO | |
| Chemical compound, drug | Sodium chloride | Sigma-Aldrich | S7653 | |
| Chemical compound, drug | Magnesium chloride hexahydrate | USP | 1374248 | |
| Chemical compound, drug | DTT | Sigma-Aldrich | D0632 | |
| Chemical compound, drug | Triton X-100 | Sigma-Aldrich | T8787 | |
| Peptide, recombinant protein | Turbo DNase I | ThermoFisher | AM2238 | |
| Other | Glass beads | Scientific Industries | SI-BG05 | |
| Other | 13 × 51 mm polycarbonate centrifuge tubes | Beckman Coulter | 349622 | |
| Chemical compound, drug | Sucrose | Sigma-Aldrich | S0389 | |
| Chemical compound, drug | Phosphate buffered saline (PBS) | Sigma-Aldrich | P4417 | |
| Commercial assay or kit | Click-iT HPG Alexa Fluor 488 Protein Synthesis Assay Kit | ThermoFisher | C10428 | |
| Chemical compound, drug | DAPI (4′,6-Diamidino-2-Phenylindole, Dihydrochloride) | ThermoFisher | D1306 | |
| Commercial assay or kit | Ribo-Zero Magnetic Gold Kit (Yeast) | Epicentre | MRZY1324 | |

*Continued on next page*

*Continued*

| Reagent type (species) or resource | Designation | Source or reference | Identifiers | Additional information |
|---|---|---|---|---|
| Commercial assay or kit | SciptSeq v2 RNA-Seq Library Preparation Kit | Epicentre | SSV21124 | |
| Antibody | Peroxidase Anti-Peroxidase (PAP) Soluble Complex | Sigma-Aldrich | P1291 | (1:1000) |
| Antibody | Anti-Pgk1p antibody, rabbit polyclonal | abcam | ab38007 | (1:1000) |
| Other | Novex WedgeWell4–12% Tris-Glycine gels | ThermoFisher | XP04125 | |
| Software, algorithm | MetaboAnalyst | https://www.metaboanalyst.ca/ | RRID:SCR_015539 | Web server for statistical, functional and integrative analysis of metabolomics data |
| Software, algorithm | AccuComp Z2 | Beckman Coulter | 383550 | Software to monitor number and size of cells with Z2 cell counter |
| Software, algorithm | NIS-Elements | https://www.nikoninstruments.com/Products/Software | RRID:SCR_014329 | Microscope imaging software suite used with Nikon products |
| Software, algorithm | ImageJ | https://imagej.net/ | RRID:SCR_003070 | Image processing software |
| Software, algorithm | Adobe Photoshop | https://www.adobe.com/products/photoshop.html | RRID:SCR_014199 | Image processing software |
| Software, algorithm | RStudio | http://www.rstudio.com/ | RRID:SCR_000432 | Software for the R statistical computing environment |
| Software, algorithm | SGD | http://www.yeastgenome.org/ | RRID:SCR_004694 | Saccharomyces Genome Database |
| Software, algorithm | R | https://www.r-project.org | v3.5.2 RRID:SCR_001905 | Statistical Computing Environment |
| Software, algorithm | PANTHER | http://www.geneontology.org/ | RRID:SCR_002811 | Gene ontology enrichment analysis |

Where known, the Research Resource Identifiers (RRIDs) are shown.

## Strains and media

All the strains used in this study are shown in the Key Resources Table, above. Unless noted otherwise, the cells were cultivated in the standard, rich, undefined medium YPD (1% $^w/_v$ yeast extract, 2% $^w/_v$ peptone, 2% $^w/_v$ dextrose), at 30°C (*Kaiser et al., 1994*). Strains (HB147 and HB171) carrying the *MET3-TAP* allele in the *rpl22aΔ*, or *rpl22bΔ*, background were generated from crosses of YSC1178-202231887 (*MET3-TAP*) with KS976 (*rpl22aΔ*), or KS979 (*rpl22bΔ*), respectively. The resulting diploids were sporulated and dissected to obtain the mutant combinations, as indicated in *Figure 5*. Similarly, we generated the double *rpl22aΔ*, *shm1Δ* (NM68); *rpl22aΔ*, *shm2Δ* (NM67); *rpl22aΔ*, *ade17Δ* (NM66) deletion mutants, shown in *Figure 7—figure supplement 1*; and the *shm2Δ*, *ade3Δ* deletion mutants (strains NM64 and NM65), shown in *Figure 8—figure supplement 1*.

## Sample-size and replicates

For sample-size estimation no explicit power analysis was used. All the replicates in every experiment shown were biological ones, from independent cultures. A minimum of three biological replicates were analyzed in each case, as indicated in the legends of each corresponding figure. For the RNAseq and Riboseq datasets, three replicates was the minimum required for the computational pipelines for RNAseq and Riboseq datasets we used, as described below. Three replicates was also the minimum required for the robust bootstrap ANOVA, which we used in the analysis of metabolite and amino acid levels (*Figure 6*), and ribosome protein abundance (*Figure 2C*), as indicated. For measurements where at least four independent replicates were analyzed, we used non-parametric statistical methods, as indicated in each case. No data or outliers were excluded from any analysis.

## SWATH-mass spectrometry

The samples used to measure ribosomal protein abundances were from the haploid strains shown in *Figure 2* (see Key Resources Table). Exponentially growing cells were quenched with 100 µg/ml cycloheximide and 0.1% sodium azide. Cells were harvested from three independent cultures of each strain (5.8E+07 cells in each sample). From the same cultures, 2.5E+07 cells for each sample were harvested for amino acid analysis (see below). For SWATH-mass spectrometry, the cells were re-suspended in standard polysome buffer (20 mM Tris·Cl (pH 7.4), 150 mM NaCl, 5 mM $MgCl_2$, 1 mM DTT, 100 µg/ml cycloheximide), containing 1% $^v/_v$ Triton X-100 and 25 U/ml Turbo DNase I, to a volume of 0.35 ml. Then, 0.2 ml of 0.5 mm glass beads were added to each sample, and vortexed at maximum speed for 15 s, eight times, placing on ice for 15 s in between. The lysates were clarified by centrifuging at 5,000 rpm for 5 m, at 4°C, and again for 5 m at 13,000 rpm at 4°C. The supernatant was transferred to a 13 × 51 mm polycarbonate ultracentrifuge tube, underlaid with 0.90 ml of 1 M sucrose, and the ribosomes were pelleted by centrifugation in a TLA100.3 rotor (Beckman) at 100,000 rpm at 4°C for 1 hr. The pellets were then re-suspended in PBS, for analysis with SWATH-mass spectrometry, as described previously (*Schilling et al., 2017*).

All the peak area measurements are in *Figure 2—source data 1*. The peak area values specifically for Rpl22Ap and Rpl22Bp used as input for *Figure 2B* are in *Figure 2—source data 1* (sheet 'RPL22_peak_areas_SWATH_MS'). The peak area values used as input for *Figure 2C* are in *Figure 2—source data 1* (sheet 'RP_peak_areas_SWATH_MS'). To identify significant differences in the comparisons among the different strains, we used the robust bootstrap ANOVA, via the *t1waybt* function, and the posthoc tests via the *mcppb20* function, of the *WRS2* R language package (*Wilcox, 2011*). For this, and all subsequent tests involving the robust bootstrap ANOVA (e.g., in the metabolite analysis), the input values for these tests were first scale-normalized so that every sample had the same total values.

## HPG incorporation, microscopy, and flow cytometry

To measure newly synthesized protein in cells (see *Figure 2—figure supplement 1*), we used the non-radioactive labeling assay kit 'Click-iT HPG Alexa Fluor Protein Synthesis Assay Kits', according to the manufacturer's instructions. Briefly, 50 µM of L-homopropargylglycine (HPG), an analog of methionine containing an alkyne moiety, was added to the exponentially growing yeast culture and incubated for 30 m at 30°C. The cells were fixed and permeabilized according to their instructions, and incubated with Alexa Fluor 488 containing an azide moiety, for 30 m in the dark, at room temperature. The alkyne and the azide groups undergo a CLICK reaction, and the incorporation of the HPG methionine analog was quantified by flow cytometry (Becton Dickinson Accuri C6). Mean fluorescence intensities were measured for 8,000–10,000 cells for each sample and then normalized by the mean fluorescence intensity of the wild type strain (*Figure 2—figure supplement 1B*).

Cells were also viewed with a Nikon Eclipse TS100 microscope, with a 100X objective, and the images were captured with a CoolSnap Dyno 4.5 Nikon camera (*Figure 2—figure supplement 1C*). The cells were also stained with DAPI, to visualize the nuclei (*Amberg et al., 2006b*). The exposure time for the DAPI, and GFP (to visualize HPG-Alexa incorporation), filters were 4 s, and 600 ms, respectively. All images were captured in NIS Elements Advanced Research (version 4.10) software. The fluorescent images acquired with the GFP and DAPI filters were processed identically in ImageJ and Adobe Photoshop.

## Riboseq and RNAseq libraries

We used the same approach we had described previously (*Blank et al., 2017*), to collect cells from elutriated cultures of each ribosomal protein deletion and generate ribosome footprint libraries. For the RNAseq libraries from the same samples, we also used the same approach we had described (*Ingolia et al., 2012*; *Blank et al., 2017*), except that we did not select for polyA-tailed RNAs. Instead, from total RNA, we depleted rRNA, using the 'Ribo-Zero Magnetic Gold Kit (Yeast)', according to the manufacturer's instructions. All libraries were sequenced on an Illumina HiSeq4000, with multiplexing, at the Texas A and M AgriLife Genomics and Bioinformatics Facility.

## Sequencing reads quality control and mapping

Raw sequencing Ribosome Profiling (RP) and Transcriptional Profiling (TP) reads (50 nucleotides, nt), were subjected to quality control as follows: First, identical reads, sequencing artifacts, and sequencing adaptors were removed. Second, reads were scanned for the presence of 'N's'. If 'N's' were found, reads were truncated at that position, and the resulting truncated fragment was discarded. The resulting retained reads were then scanned for nucleotides whose Quality Score (Q-score) was 19 or lower. When and if found, the read in question was then truncated at that position and the resulting truncated downstream fragment was discarded. The corresponding retained fragment was then evaluated for length. Reads that were 25 nt or longer (for TP-reads) or 15 nt or longer (for RP-reads), were retained for analysis.

Reads that passed quality control were mapped to the coding exons of the *Saccharomyces cerevisiae* genome (R64-1-1 (GCA_000146045.2)), using a combination of Bowtie-directed alignment (Bowtie version 1.2.2) (*Langmead et al., 2009*; *Trapnell et al., 2013*). Resulting BAM files were then processed by RSEM-Calculate-Expression (https://github.com/deweylab/RSEM), which provides a posterior mean and 95% credibility interval estimates for expression levels (*Li and Dewey, 2011*). When needed, resulting alignment BAM files were sorted and indexed using the SAM Tools (*Li, 2011a*; *Li, 2011b*; *Li et al., 2009*) and visualized using Integrative Genome Browser (IGV) (*Robinson et al., 2011*; *Thorvaldsdóttir et al., 2013*).

Using the same mapping pipeline, we re-mapped the reads from our previous study with wild type cells (*Blank et al., 2017*). The values for the raw reads from all the strains and cell cycle points are in *Figure 3—source data 1*. The corresponding Transcripts Per Kilobase Million (TPM) values are in *Figure 3—source data 2*. Raw sequencing data from each library have been deposited (GEO: GSE135336).

## Computational pipelines for RNAseq and riboseq

To identify transcripts whose abundance was significantly different between any two strains at any of the eight different points in the cell cycle we evaluated, we used the R language packages *babel* (*Olshen et al., 2013*) and *DESeq2* (*Love et al., 2014*). For both packages, the raw read data (*Figure 3—source data 1*) were used as input. In *babel*, the number of permutations was set to ten million (nreps = 1e+07), so the minimum p-value is 1/nreps, and the minimum read count for a transcript to be included in the analysis was set to 20 ('min.rna=20'). All mRNAs with significantly different abundance between two strains, at any one point in the cell cycle, had an adjusted p-value or false discovery rate (FDR) of <0.05 in *both* analyses, and a fold-change ≥4, and they are shown in *Figure 4—source data 1*.

To identify transcripts with significantly altered translational efficiency, we used the R language packages *babel* and *Riborex* (*Li et al., 2017*), the last run with the *DESeq2* and with the *edgeR* (*Dai et al., 2014*) platforms. Again, for all packages, the raw read data (*Figure 3—source data 1*) were used as input, and the minimum read count for a transcript to be included in the analysis was set to 20 ('min.rna=20' in *babel*, and 'mimiminMeanCount = 20' in *Riborex*). All mRNAs with significantly different translational efficiency between two strains, at any one point in the cell cycle, had an adjusted p-value or false discovery rate (FDR) of <0.05 in *all* analyses, and a fold-change ≥2, and they are shown in *Figure 5—source data 1*.

## Immunoblot analysis

For protein surveillance, protein extracts were made as described previously (*Amberg et al., 2006a*), and run on 4–12% Tris-Glycine SDS-PAGE gels. To detect TAP-tagged Met3p with the PAP

reagent, we used immunoblots from extracts of the indicated strains as we described previously (*Blank et al., 2017*). Loading was evaluated with an anti-Pgk1p antibody. In each experiments several technical, repeat samples were loaded on each gel (see *Figure 5*).

### Cell size and DNA content measurements

The methods to measure the cell size of asynchronous cultures and estimate the critical size of asynchronous cultures, have been described in detail previously (*Guo et al., 2004*; *Maitra et al., 2019*; *Soma et al., 2014*; *Truong et al., 2013*). DNA content was measured by flow cytometry as we described previously (*Hoose et al., 2012*).

### Metabolic profiling

The untargeted, primary metabolite and biogenic amine analyses were done at the NIH-funded West Coast Metabolomics Center at the University of California at Davis, according to their established mass spectrometry protocols. Extract preparation was also done at the same facility, from 1E+07 cells in each sample. The cells were collected from exponentially growing cultures, quenched with 100 µg/ml cycloheximide and 0.1% sodium azide. Cells were harvested from three independent cultures of each strain (1E+07 cells in each sample), washed in water containing the same concentrations of cycloheximide and azide, and provided to the Metabolomics facility as frozen (at −80℃) pellets. To identify significant differences in the comparisons among the different strains, we used the robust bootstrap ANOVA, via the *t1waybt* function, and the posthoc tests via the *mcppb20* function, of the *WRS2* R language package (*Wilcox, 2011*). The input values we used, after they were scaled-normalized for input intensities per sample, are in *Figure 6—source data 1*. For the primary metabolites, the data are in *Figure 6—source data 1* (sheet 'pm_input'). For the biogenic amines, the data are in *Figure 6—source data 1* (sheet 'ba_input'). Detected species that could not be assigned to any compound were excluded from the analysis.

### Amino acid analysis

Samples for amino acid analysis were prepared as we described previously (*He et al., 2014*), from the same cultures used for the SWATH-mass spectrometry experiments (see above), using 2.5E+07 cells for each sample. The PTH-based amino acid analyses were done at the Texas A and M Protein Chemistry Facility. Statistical tests for significant differences between the different strains were done as described above for the other metabolites. The input values we used (in nmoles), after they were normalized for total counts per sample, are in *Figure 6—source data 1* (sheet 'aa_input').

### Replicative lifespan assays

All replicative lifespan assays were done on solid YPD medium, as described previously (*Steffen et al., 2009*).

## Acknowledgements

This work was supported by grant GM123139 from the National Institutes of Health to MP, BKK, and RAR, and by grant P30 AG013280 to MK. We also acknowledge the support from the NCRR shared instrumentation grant 1S10 OD016281 (Buck Institute) and from NIH grant 1U24DK097154 (UC Davis 'West Coast Metabolomics Center').

## Additional information

#### Competing interests

Matt Kaeberlein: Reviewing editor, *eLife*. The other authors declare that no competing interests exist.

## Funding

| Funder | Grant reference number | Author |
|---|---|---|
| National Institutes of Health | GM123139 | Rodolfo Aramayo<br>Brian K Kennedy<br>Michael Polymenis |
| National Center for Research Resources | 1S10 OD016281 | Birgit Schilling |
| National Institute on Aging | P30 AG013280 | Matt Kaeberlein |

The funders had no role in study design, data collection and interpretation, or the decision to submit the work for publication.

## Author contributions

Nairita Maitra, Chong He, Heidi M Blank, Formal analysis, Investigation, Writing - review and editing; Mitsuhiro Tsuchiya, Formal analysis, Investigation; Birgit Schilling, Resources, Data curation, Formal analysis, Investigation, Writing - review and editing; Matt Kaeberlein, Resources, Formal analysis, Supervision, Methodology, Writing - review and editing; Rodolfo Aramayo, Resources, Data curation, Software, Formal analysis, Funding acquisition, Methodology; Brian K Kennedy, Conceptualization, Resources, Supervision, Funding acquisition, Methodology, Writing - review and editing; Michael Polymenis, Conceptualization, Resources, Data curation, Formal analysis, Supervision, Funding acquisition, Validation, Investigation, Visualization, Methodology, Writing - original draft, Project administration, Writing - review and editing

## Author ORCIDs

Matt Kaeberlein (iD) http://orcid.org/0000-0002-1311-3421
Rodolfo Aramayo (iD) http://orcid.org/0000-0001-9702-6204
Michael Polymenis (iD) https://orcid.org/0000-0003-1507-0936

## Decision letter and Author response

Decision letter https://doi.org/10.7554/eLife.53127.sa1
Author response https://doi.org/10.7554/eLife.53127.sa2

# Additional files

## Supplementary files

• Transparent reporting form

## Data availability

Sequencing data have been deposited in GEO under accession code GSE135336. All data generated or analysed during this study are included in the manuscript and supporting files.

The following dataset was generated:

| Author(s) | Year | Dataset title | Dataset URL | Database and Identifier |
|---|---|---|---|---|
| Aramayo R, Polymenis M | 2019 | Paralog-specific phenotypes of ribosomal protein mutants identify translational control mechanisms in the cell cycle and replicative longevity | https://www.ncbi.nlm.nih.gov/geo/query/acc.cgi?acc=GSE135336 | NCBI Gene Expression Omnibus, GSE135336 |

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
