## [Decision Letter]

**Acceptance summary:**

This study provides an interesting explanation for why yeast cells lacking specific ribosomal proteins, in this case Rpl22a, have a longer lifespan. The resulting reduction in one-carbon metabolism pathway, coupled with the longer replicative lifespan in cells defective for this pathway, suggests it could be a potential target for longevity interventions.

**Decision letter after peer review:**

Thank you for submitting your article "Translational control of one-carbon metabolism underpins ribosomal protein phenotypes in cell division and longevity" for consideration by *eLife*. Your article has been reviewed by three peer reviewers, and the evaluation has been overseen by a Reviewing Editors and by Jessica Tyler as the Senior Editor.

The reviewers have discussed the reviews with one another and the Reviewing Editor has drafted this decision to help you prepare a revised submission.

Summary:

This paper follows up on interesting findings that certain deletion mutants of individual ribosomal protein genes extend replicative lifespan of *Saccharomyces cerevisiae*, while others do not. They focus on the RPL22A/B paralogous gene pair, where rpl22a∆ mutants are long lived compared to WT and the rpl22b∆ mutant. Their goal was to determine the underlying mechanism of RLS extension induced by the rpl22a∆ paralog mutation. They clearly show that longevity does not correlate with doubling time, ribosome abundance or content, or critical cell size. Using carefully done riboseq analysis, the authors demonstrate downregulated translation of mRNAs related to 1C metabolism in the rpl22a∆ mutant. The reduced 1C metabolism was also supported by excellent metabolite profiling. Together, these results support the hypothesis that extended RLS in the absence of the Rpl22a paralog is correlated with reduced activity of the 1C pathway. However, the results on directly tying reduced 1C activity as causative to lifespan regulation in the rpl22a∆ mutant are not fully convincing at this time.

Essential revisions:

The reviewers identified a number of concerns that are listed below, and should be addressed through experimentation and/or rewriting where appropriate. One of the key issues was that authors are using comparisons of certain pairs of strains to make conclusions about the differences between other pairs, rather than doing direct comparisons. Streamlining the presentation is essential and should help clarify some the specific concerns.

1) Since rpl22a deletion influences many aspects of cell physiology, it is not clear whether the lifespan-extending effect of rpl22a deletion is completely through the repression of 1-C pathways, or it is just one of the contributing factors. I would suggest additional experiments to examine the lifespans of double mutants, such as rpl22aΔ, shm1Δ and rpl22aΔ, ade1Δ. If the double mutants can further extend lifespan compared to rpl22aΔ alone, a discussion of this result would be needed.

2) In Figure 6, the effects of shm1∆ and shm2∆ deletion mutants on RLS are modest, with the shm2 mutant only separating from BY4742 after 35 days. There is no statistical analysis provided, or an indication of whether the results were reproducible. This is a key result arguing that inhibiting 1C metabolism pathways extend RLS, so it is critical to convincingly demonstrate this.

3) The ade17∆ mutant also extended RLS and is included as another example of a 1C mutant. However, the Ade17 enzyme is actually part of the de novo purine biosynthesis pathway. It is therefore surprising that another de novo pathway mutant, ade2∆, had no effect on RLS. How can these disparate results be reconciled?

4) If 1C metabolism is truly mediating the effects of the rpl22a deletion, then supplementing with 1C inputs and products could potentially suppress the long RLS of the rpl22a mutant, thus supporting the overall hypothesis of the paper.

5) In Figure 8—figure supplement 1, SHM2 translation efficiency is reduced in the rpl22a/rpl22a diploid mutant as expected from earlier figures, but now it looks like the translation efficiency is also reduced in the rpl22b/rpl22b diploid mutant, which does not have extended lifespan. This result does not seem to support the overall conclusion of specificity for the rpl22a mutant. How do the authors interpret this result? Minimally, a better explanation of the data and conclusion are needed.

6) Related to this point, while the translational differences between rpl22aΔ and rpl22bΔ are clearly presented in Figure 4, the differences between the mutants and WT (Figure 5—figure supplement 3) are very hard to follow due to the quality of presentation. Please enlarge Figure 5—figure supplement 3 and make the gene names readable. This figure might be quite important to support the claim about quantitative differences. Also for the same point, a WT control would be needed for Figure 4C. It is clear that Met3 expression is lower in rpl22aΔ than that in rpl22bΔ, but it would be good to know how those compare to the level in WT.

7) In Figure 8, there is clearly shown an increased cell size for an shm2∆ ade3∆ double mutant, without any significant change in DNA content, suggesting that loss of these two 1C enzymes impinges on multiple cell cycle phases. However, this is not linked back with lifespan regulation. Do shm1∆ or shm2∆ mutants also affect multiple cell cycle phases? Does this double mutant, or an ade3∆ single mutant extend RLS? The ade3∆ mutant is especially relevant because this mutation impacts both 1C and de novo purine biosynthesis, unlike the ade2 and ade17 mutants in Figure 6.

8) The authors also investigate differences between the rpl34aΔ and rpl34bΔ deletion mutants (comparing them to WT as well), which increases complexity of the presentation without too much additional benefit. If I understand correctly, the Rpl34 strains are used as a kind of "control" for the RNAseq and TE experiments to verify that the assays pick up real differences between rpl22aΔ and rpl22bΔ. However, the authors also compare the Rpl34 deletion strains to WT and make remarks about them, which further complicate the whole picture and generate more questions than they answer. For example, can we conclude that 1C metabolism underlies the increased longevity of the Rpl34 mutants relative to WT? What is the importance of the differences between the Rpl22 and Rpl34 mutants described on Figure 3B and Figure 5—figure supplement 1B?

9) The authors briefly comment that the rpl22a,bΔ strain is not long-lived (subsection “Loss of Rpl22Ap reduces overall protein synthesis”), but do not explore this observation further. What should be concluded out of this comparison? Is 1C metabolism similarly perturbed in the double deletion strain, or is Rpl22b deletion somehow "reversing" the effect of the lack of Rpl22a? What is the RLS of rpl22a,bΔ and how does it compare to WT?

10) Regarding a technical aspect of the work: the authors conclude that the synchrony of their elutriated cultures is good (subsection “Generating RNAseq and Riboseq libraries from synchronous, dividing cells lacking ribosomal protein paralogs”, second paragraph). However, Figure 2B shows that the budding index is starting to increase a bit too fast (compare Figure 1B of Blank et a., 2017). Also, Clb2 seems to ramp up a bit too early, and I cannot really detect a peak in Hhf1 expression. However, my main concern with the elutriation data is related to what the authors state: cells with the same volume across all mutants may very well be in different cell cycle phases (for example, G1 of rpl22aΔ is clearly longer than the G1 of the other strains). The authors compare RNAseq and TE measurements of different strains across different volumes and report genes that differ in expression even at a single time point. Yet, given the approach they follow, some of the differences they uncover could be simply caused by this shift in cell cycle phases rather than differences in expression at the same cell cycle phase. How can one make sure that the reported differences really reflect changes in cell cycle expression and not just differences in timing? Perhaps the authors could show more time series such as the one of Figure 8—figure supplement 1 (e.g. for Ade17, Shm1, Met3) to demonstrate the differences across mutants in a visually clear manner.

11) On which criteria was the selection of 1C enzyme deletions (subsection “Genetic interventions in 1C metabolism that extend longevity”) made? What is their effect on RLS compared to MET3? (i.e. what is the increase in RLS caused by MET3 deletion? Are the numbers comparable?). Does metabolomics support the fact that these deletions extend RLS through 1C metabolism?

---

## [Author Response]

Essential revisions:The reviewers identified a number of concerns that are listed below, and should be addressed through experimentation and/or rewriting where appropriate. One of the key issues was that authors are using comparisons of certain pairs of strains to make conclusions about the differences between other pairs, rather than doing direct comparisons. Streamlining the presentation is essential and should help clarify some the specific concerns.1) Since rpl22a deletion influences many aspects of cell physiology, it is not clear whether the lifespan-extending effect of rpl22a deletion is completely through the repression of 1-C pathways, or it is just one of the contributing factors. I would suggest additional experiments to examine the lifespans of double mutants, such as rpl22aΔ, shm1Δ and rpl22aΔ, ade1Δ. If the double mutants can further extend lifespan compared to rpl22aΔ alone, a discussion of this result would be needed.

We did this experiment, and show it in a new supplementary figure (Figure 7—figure supplement 1). We describe these results in the text:

“To test the extent to which single deletions of 1C enzymes could contribute to longevity independently of rpl22aΔ cells, we measured the longevity of rpl22aΔ cells in the context of ade17Δ, shm1Δ, or shm2Δ deletions (Figure 7—figure supplement 1). […] We conclude that loss of 1C enzymes does not further extend the longevity of rpl22aΔ cells.”

2) In Figure 6, the effects of shm1∆ and shm2∆ deletion mutants on RLS are modest, with the shm2 mutant only separating from BY4742 after 35 days. There is no statistical analysis provided, or an indication of whether the results were reproducible. This is a key result arguing that inhibiting 1C metabolism pathways extend RLS, so it is critical to convincingly demonstrate this.

The results are significant (in every case p<0.0001, based on the log-rank test; we added this in the legend of the figure). We added results for ade3 cells as well, as requested in a subsequent point (see revised Figure 7). As further support for the reproducibility of these effects, an independent analysis of the single mutants was performed along with double mutants combined with rpl22aΔ and lifespan extension was significant in each case for the single mutants (see Figure 7—figure supplement 1).

3) The ade17∆ mutant also extended RLS and is included as another example of a 1C mutant. However, the Ade17 enzyme is actually part of the de novo purine biosynthesis pathway. It is therefore surprising that another de novo pathway mutant, ade2∆, had no effect on RLS. How can these disparate results be reconciled?

The parts of the pathway are not the same. Ade2p catalyzes a step in purine synthesis, but it is not part of one-carbon metabolism. Ade17p directly transfers 1C from folate (formyl-THF) to make IMP. Ade2p-mediated carboxylation does not involve folates. We clarified this distinction, which supports the notion that 1C, folate-based metabolism plays a key role:

“Interestingly, all three enzymes encoded by these genes, including Ade17p with its 5-aminoimidazole-4-carboxamide ribonucleotide transformylase activity, catalyze folate-dependent reactions (Tibbetts and Appling, 2000). In contrast, the carboxylase activity of Ade2p is not folate-dependent.”

4) If 1C metabolism is truly mediating the effects of the rpl22a deletion, then supplementing with 1C inputs and products could potentially suppress the long RLS of the rpl22a mutant, thus supporting the overall hypothesis of the paper.

We considered this approach, however, the multiple outputs and requirements of 1C metabolism cannot be corrected with simple additions of exogenous metabolites. As shown in Figure 6—figure supplement 1, supplementation with Ser, Gly, Ade, Met and combinations did not suppress the growth defect of rpl22a cells, making it unlikely that they would suppress the longevity of rpl22a cells. In contrast, loss-of-function mutations in 1C enzymes recapitulate the coordinate down-regulation of 1C metabolism in rpl22 cells and extend longevity (Figure 7). The additional experiments, suggested by the reviewers (see points 1 and 2), further support the overall hypothesis that down-regulation of 1C enzymes is responsible, at least in part, for the longer lifespan of rpl22a cells.

5) In Figure 8—figure supplement 1, SHM2 translation efficiency is reduced in the rpl22a/rpl22a diploid mutant as expected from earlier figures, but now it looks like the translation efficiency is also reduced in the rpl22b/rpl22b diploid mutant, which does not have extended lifespan. This result does not seem to support the overall conclusion of specificity for the rpl22a mutant. How do the authors interpret this result? Minimally, a better explanation of the data and conclusion are needed.

We interpret these results to be consistent with our model. The confusion arises because in the (now re-numbered) Figure 8—figure supplement 1 we do not compare translational efficiencies between strains, only within each strain, at different points in the cell cycle. We added the following, to clarify this point: “Note that for the data shown in Figure 8—figure supplement 1, the translational efficiency of SHM2 is normalized against all the points in the cell cycle within each strain, not across the different strains.”

6) Related to this point, while the translational differences between rpl22aΔ and rpl22bΔ are clearly presented in Figure 4, the differences between the mutants and WT (Figure 5—figure supplement 3) are very hard to follow due to the quality of presentation. Please enlarge Figure 5—figure supplement 3 and make the gene names readable. This figure might be quite important to support the claim about quantitative differences. Also for the same point, a WT control would be needed for Figure 4C. It is clear that Met3 expression is lower in rpl22aΔ than that in rpl22bΔ, but it would be good to know how those compare to the level in WT.

A clearer overview with all pair-wise comparisons is in Figure 5—figure supplement 1. Regarding Figure 5—figure supplement 3, while we understand the reviewer’s concern, we enlarged it somewhat, but it cannot be enlarged further. Note that we also provide all the gene names in Figure 5—source data 1. This is now mentioned in the Materials and methods and also in the legend of the figures to aid readers.

Lastly, we now added the WT control in Figure 5C. It is clear that in rpl22a cells Met3 levels are lower than in WT and in rpl22b cells. Interestingly, Met3 levels are higher in rpl22b than in WT cells.

7) In Figure 8, there is clearly shown an increased cell size for an shm2∆ ade3∆ double mutant, without any significant change in DNA content, suggesting that loss of these two 1C enzymes impinges on multiple cell cycle phases. However, this is not linked back with lifespan regulation. Do shm1∆ or shm2∆ mutants also affect multiple cell cycle phases?

We made these measurements and added the following:

“We note that the cell cycle profile of wild type cells (33% of the cells in G1; Figure 8C) is not affected at all by the loss of Ade3p or Shm2p (32-33% G1 cells Figure 8C), and only slightly so by the loss of Shm1p or Ade17p (27-28% G1 cells; not shown). […] It is also mild, but not severe delays in cell cycle progression that in some cases extend longevity (He et al., 2014).”

Does this double mutant, or an ade3∆ single mutant extend RLS?

As we noted above, the double *shm1Δ,shm2Δ* mutant is lethal, and yes, *ade3Δ* alone does extend longevity (the new data was added in Figure 7).

The ade3∆ mutant is especially relevant because this mutation impacts both 1C and de novo purine biosynthesis, unlike the ade2 and ade17 mutants in Figure 6.

See our comments and the new data we mentioned above. We also note that *ade17Δ* also impacts both 1C and de novo purine biosynthesis, and it does extend lifespan (Figure 7C).

8) The authors also investigate differences between the rpl34aΔ and rpl34bΔ deletion mutants (comparing them to WT as well), which increases complexity of the presentation without too much additional benefit. If I understand correctly, the Rpl34 strains are used as a kind of "control" for the RNAseq and TE experiments to verify that the assays pick up real differences between rpl22aΔ and rpl22bΔ. However, the authors also compare the Rpl34 deletion strains to WT and make remarks about them, which further complicate the whole picture and generate more questions than they answer. For example, can we conclude that 1C metabolism underlies the increased longevity of the Rpl34 mutants relative to WT? What is the importance of the differences between the Rpl22 and Rpl34 mutants described on Figure 3B and Figure 5—figure supplement 1B?

We apologize for the confusion and have attempted to clarify the rationale for this analysis. The benefit and value of analyzing rpl34 mutants is that this is another paralog pair, never examined before. When we started out, the rpl34 mutants were not considered a control. Nonetheless, the data show that their transcriptomic and translatomic differences are very similar. Given that they are both long-lived, this is interesting, unlike the rpl22 pair which differ phenotypically and in the various molecular profiling assays we showed. It certainly does not contradict, and rather supports, our conclusion that longevity is associated with altered translational efficiency of specific mRNAs (we added this statement in the last paragraph of the subsection “Transcripts with altered relative translational efficiency in rpl22 and rpl34 mutants”).

As for the various pairwise comparisons, we understand that it might be overwhelming, but we erred on the side of completeness. It is also valuable information. To clarify further, regarding the rpl34 mutant, we now added the following: “The down-regulation of mRNAs encoding 1C enzymes in rpl34aΔ cells is consistent with the hypothesis that down-regulation of 1C metabolism is associated with increased longevity, as we will describe later in the manuscript.”

As for the rpl22 vs. rpl34 comparisons, aside from the value of showing a complete analysis, the most frequent significant changes are in ‘cytoplasmic translation’. Again, given that we are dealing with ribosomal protein mutants this may not seem surprising, but it is one thing to expect something and quite another to actually show it.

9) The authors briefly comment that the rpl22a,bΔ strain is not long-lived (subsection “Loss of Rpl22Ap reduces overall protein synthesis”), but do not explore this observation further. What should be concluded out of this comparison? Is 1C metabolism similarly perturbed in the double deletion strain, or is Rpl22b deletion somehow "reversing" the effect of the lack of Rpl22a? What is the RLS of rpl22a,bΔ and how does it compare to WT?

All the RLS data for the double rpl22 mutant (not long-lived) have been reported (in Steffen et al., 2012), as we cite. Here, we only used rpl22ab cells in our measurements of ribosome composition (Figure 2) and protein synthesis rates (Figure 2—figure supplement 1). The point Figure 2 makes, is that Rpl22 perturbations, including the complete lack of it in rpl22ab cells, does not change ribosome composition, as we state: “In all strains tested, including the rpl22a,bΔ cells lacking Rpl22 altogether, the relative proportion of the RPs in ribosomes was essentially constant, indicated by the Spearman rank correlation coefficients (ρ), which were very high (≥0.95) in each case.”

Likewise, for the protein synthesis measurements we showed (Figure 2—figure supplement 1), we state: “Because rpl22a,bΔ cells are not long-lived (Steffen et al., 2012), we conclude that merely reducing rates of protein synthesis is not sufficient to promote longevity. This conclusion is in agreement with the observation that inhibiting protein synthesis with cycloheximide also does not increase lifespan (Steffen et al., 2008).”

We did not do ribosome profiling of rpl22ab cells, and we cannot comment on any specific pathways, including 1C metabolism, in these cells. However, even if we had done those extensive experiments we wouldn’t know how to interpret them with regards to longevity. The double mutant has a strong growth defect (proliferates slower than rpl22a cells, see Figure 2). Even if 1C metabolism is down-regulated in these cells, it is not clear to us how one can draw conclusions about longer lifespan from a mutant that grows poorly and it is not long-lived. Instead, all the signatures we report implicating 1C metabolism were made from long-lived mutants.

10) Regarding a technical aspect of the work: the authors conclude that the synchrony of their elutriated cultures is good (subsection “Generating RNAseq and Riboseq libraries from synchronous, dividing cells lacking ribosomal protein paralogs”, second paragraph). However, Figure 2B shows that the budding index is starting to increase a bit too fast (compare Figure 1B of Blank et al., 2017). Also, Clb2 seems to ramp up a bit too early, and I cannot really detect a peak in Hhf1 expression. However, my main concern with the elutriation data is related to what the authors state: cells with the same volume across all mutants may very well be in different cell cycle phases (for example, G1 of rpl22aΔ is clearly longer than the G1 of the other strains). The authors compare RNAseq and TE measurements of different strains across different volumes and report genes that differ in expression even at a single time point. Yet, given the approach they follow, some of the differences they uncover could be simply caused by this shift in cell cycle phases rather than differences in expression at the same cell cycle phase. How can one make sure that the reported differences really reflect changes in cell cycle expression and not just differences in timing? Perhaps the authors could show more time series such as the one of Figure 8—figure supplement 1 (e.g. for Ade17, Shm1, Met3) to demonstrate the differences across mutants in a visually clear manner.

Unfortunately, for the ribosome profiling experiments it is not possible to extrapolate “time-points” from cell size points, due to different strains with different doubling time. It can also lead to artifacts when elutriated cultures are analyzed, even if we were dealing with one strain. All elutriations, even when aliquots are followed from the same elutriated sample at different time points (we present some of those, see Figure 8), should always be normalized on the basis of size. Specifically for the figure the reviewer comments on, we have done our ribosome profiling from cell-size series, from pools of cells collected at the same cell size. That is the only way we could obtain enough cells to do our ribosome profiling, as we had done in our prior study (Blank et al., 2017). Furthermore, the different series for the different strains are directly comparable to each other on the basis of size. For a study that seeks to uncover growth-related effects (through protein synthesis), we view this as an advantage. Lastly, regarding the magnitude of the increase in cell cycle transcripts (e.g., for histones), we note that the scale of the axis is in Log2 values. We also kept the scale the same for all panels. This may make the magnitude of the effect to appear smaller in histones vs. CLB2, but is still evident when one compares histones vs. actin. For these reasons, we respectfully disagree that this is a concern.

11) On which criteria was the selection of 1C enzyme deletions (subsection “Genetic interventions in 1C metabolism that extend longevity”) made?

We now added the following: “ADE17 and SHM2 encode 1C enzymes and they were translationally down-regulated in rpl22aΔ cells. ADE2 does not encode a 1C enzyme and its translational efficiency did not change in rpl22aΔ vs. rpl22bΔ cells. We also tested shm1Δ cells, because even though the translational efficiency of SHM1 was similar in rpl22aΔ vs. rpl22bΔ cells, Shm1p catalyzes in mitochondria the same reaction that Shm2p does in the cytoplasm.”

What is their effect on RLS compared to MET3? (i.e. what is the increase in RLS caused by MET3 deletion? Are the numbers comparable?).

We thank the reviewer for this suggestion. Yes, the lifespan extension upon loss of Met3p is ~15%, and similar to the magnitude of the lifespan extension in shm1, shm2 or ade17 deletions. We added the following: “We note that loss of Met3p leads to a similar magnitude (≈15%) of lifespan extension (McCormick et al., 2015).”

Does metabolomics support the fact that these deletions extend RLS through 1C metabolism?

There is literature reporting various metabolic perturbations of 1C metabolites, but that is not surprising given that the corresponding enzymes (Shm1, Shm2, Ade17) are core players of 1C metabolism. We are not sure how to interpret the reviewer’s question whether they ‘support the fact that these deletions extend RLS through 1C metabolism’. We also note that metabolomic analyses of log phase cells must be interpreted extremely cautiously when considering potential effects on replicative aging, as old cells represent an extremely small fraction (roughly 1/(2^n) where n represents age) of a log phase population.